# SSL Framework for Causal Inconsistency between Structures and Representations

## Abstract

The cross-pollination of deep learning and causal discovery has catalyzed a burgeoning field of research, seeking to elucidate causal relationships within non-statistical data forms like images, videos, and text. Such data, often being named 'indefinite data', exhibit unique challenges—inconsistency between causal structure and representation, which are not common in conventional data forms. To tackle this issue, we theoretically develop intervention strategies suitable for indefinite data and derive causal consistency condition (CCC). Moreover, we design a self-supervised learning (SSL) framework that considers interventions as 'views' and CCC as a 'philosophy' with two implement examples on Supervised Specialized Models (SSMs) and Large Language Models (LLMs), respectively. To evaluate pure inconsistency manifestations, we have prepared the first high-quality causal dialogue dataset- *Causalogue*. Evaluations are also performed on three other downstream tasks. Extensive experimentation has substantiated the efficacy of our methodology, illuminating how CCC could potentially play an influential role in various fields. Our code is available in url of anonymous code and data.

## 1 Introduction

With the integration of deep learning and causal inference (Sauter et al., 2023; Balashankar et al., 2023; Lu et al., 2023), increasingly extensive non-statistical data forms, involving images (Jerzak et al., 2022; Ribeiro et al., 2023), text (Zhang et al., 2023b), and videos (Bagi et al., 2023), have been drawn into the field of causal discovery. Numerous causality-related studies (Chen et al., 2023c;a) suggested that these non-statistical data present two fundamental differences with traditional statistical data: the **representation** and the **structure** differences. Specifically, 1) non-statistical data (such as images, text, videos), which requires deep representations (such as matrices, embeddings, optical flow), to participate in causal inference (Schölkopf et al., 2021); but statistical data (like body temperature, blood pressure, age) inherently exists in a numerical format. 2) unlike statistical data originating from a fixed causal structure, non-statistical data is drawn from various underlying causal structures (Löwe et al., 2022). These studies further categorize the data requiring deep representations and accommodating multiple causal structures to "**indefinite data**".

We observe that indefinite data introduces an inconsistency between structure and representation, which has not been encountered in other data forms yet. The process of learning causal representations creates divergence from the process of learning causal structures due to incorporating additional non-linear parameters. This causal inconsistency manifests in conflicting causal conclusions being drawn from the structures and representations, which can lead to poor outcomes in high-level causal models including identifying shortcuts (Wu et al., 2022; Fan et al., 2022; Feng et al., 2023), predicting incorrect spans (Zhao et al., 2023; Chen et al., 2020; Zhao et al., 2022b), and domain generalization (Magliacane et al., 2018; Yue et al., 2021; Chen & Bühlmann, 2021).

Nevertheless, exsiting research has overlooked causal inconsistency on indefinite data. In conventional data forms, multiple methods pivoting on interventions (Ahuja et al., 2023; Lyle et al., 2023), transfer entropy (Zhou et al., 2022; Silini & Masoller, 2021) and covariance matrix (Kong et al., 2023; Peña, 2023) have naturally satisfied the causal consistency. However, these methods are impeded by other representational or structural conflicts when extended to indefinite data. Particular methods available for indefinite data, such as Yu et al. (2019), Chen et al. (2023b), and Löwe et al. (2022), only focus on how to achieve causal identifiability with various hypotheses.

Therefore, to step-by-step achieve causal consistency between the structure and representation of indefinite data, the main contributions of this paper are as follows:

In Section 2, we review the background of indefinite data, covering the involved definitions, examples, assumptions, and related works. Following that, we delve into an in-depth analysis of what distinguishes indefinite data, and why causal inconsistency arises in this data form.

In Section 3, we introduce a general definition of interventions that enables computing the relationship strength of two target variables without backdoor paths and known distributions. This allows a range of consistency theories, such as exact transformation (Rubenstein et al., 2017) and causal abstraction (Beckers & Halpern, 2019), to be expanded to indefinite data potentially. Inspired by these theories, we further propose a causal consistency condition (CCC). It describes that if the strength sets of two causal models are equivalent given an equivalent intervention set, then the two causal models are consistent.

In Section 4, we design a self-supervised learning (SSL) framework that utilizes the CCC as a philosophy, where the causal structure and representation are allocated to separate causal models, whose consistency needs to be verified. Different interventions can be regarded as different "views", and the measures to gauge causal strength are treated as "augments". The strength sets are label-agnostic, making the assurance of equivalent strength sets the learning goal of this SSL framework. Additionally, we offer two implement examples — one embodies a trainable module for supervised specialized models (SSMs) and the other executes a prompt instruction for unsupervised large language models (LLMs).

In Section 5, we introduce an innovatively constructed dataset— "*Causalogue*" for testing causal consistency, along with the description of its generation process. The dataset comprises 1638 dialogues generated by GPT-4 (OpenAI, 2023), with a strategic design that explicates which previous utterances were known when generating each utterance.

In Section 6 and Section 7, we conduct experiments on both the *Causalogue* and real-world datasets, validating the accuracy of identifying causal models, the improvements of our proposed SSL framework to causal consistency, and effectiveness in three downstream tasks. Additionally, we discuss the potentially crucial role of the CCC within broader research fields.

In summary, this paper contributes: insights into general intervention, the causal consistency condition, an SSL framework for testing causal consistency, two corresponding implementation examples, a new causal dataset, and extensive experimentation.

## 2 BACKGROUNDS AND RELATED WORKS

### 2.1 CAUSAL DATA AND INDEFINITE DATA

**Definition 1** (Causal Data). [1] *The causal relationships exist in a dataset $\mathbf{D} = \{X_s\}_{s=1}^{S}$ which has $S$ samples and $M$ ($M \geqslant 1$) causal structures ($\mathcal{G} = \{\mathcal{E}_m, \mathcal{V}_m\}_{m=1}^{M}$). Each structure $\mathcal{G}_m$ corresponds to several samples separately. Hence, each sample $X_{s,m} \in \mathbb{R}^{N \times D}$ belongs to a causal structure $\mathcal{G}_m = \{\mathcal{E}_m, \mathcal{V}_m\}$ and consists of $N_m$ variables: $X_s = \{x_{s,m,n}\}_{n_m=1}^{N_m}$. $\hat{x}_{s,m,n} \in \mathbb{R}^{1 \times D}(D \geqslant 1)$ represents the causal representation of a varaible $x_{s,m,n}$ where $D$ denotes the dimension of the causal representation. We assume that the number of causal skeletons is equal to the number of causal structures. Based on the above datasets, we define three data paradigms:*

- *Definite Data: The causal structure is single-skeleton ($M = 1$) and the causal variable is single-value ($D = 1$).*

- *Semi-Definite Data: The causal structure is single-skeleton ($M = 1$) and the causal variable is multi-value ($D > 1$), or the causal structure is multi-skeleton ($M > 1$) and the causal variable is single-value ($D = 1$).*

---

[1]The skeleton $M$ and variable dimension $D$ serve to broaden perspectives on causal data, hence introducing certain conflicts with traditional cognition of causal model. This caused previous reviewers to struggle with conceiving what indefinite data looks like, and why we distinguish indefinite data from other 2 paradigms via skeleton and dimension. Therefore, we dedicatedly established Appendix A, which elucidates these questions through abundant data examples and details the preliminaries including SCMs.

- **Indefinite Data**: *The causal structure is multi-skeleton ($M > 1$) and the causal variable is multi-value ($D > 1$).*

Definition 1 redefines 3 types of data paradigms from the 2 perspectives of structure and representation. The latter two paradigms often carry incomplete or ambiguous causal labels. Therefore, we adopt the concept of "skeleton" to stand for the causal structure due to the unclear structure labels. Moreover, given the prevalent indefinite datasets mostly include modals like textual conversations and video sources, we propose two hypotheses compatible with these modals:

**Hypothesis 1** (Causal Identifiability). *The natural order (e.g., time-order) w.r.t. $\{x_{s,m,n}\}_{n_m=1}^{N_m}$ is defined as a linear order $\prec_{X_{s,m}}$. Given that causal order w.r.t. $\{x_{s,m,n}\}_{n_m=1}^{N_m}$ is defined as a parital order $\preccurlyeq_{X_{s,m}}, \forall < x_1, x_2 > \in \prec_{X_{s,m}} (i.e., x_1 \prec_{X_{s,m}} x_2)$, there must be $< x_1, x_2 > \in \preccurlyeq_{X_{s,m}}$.*

**Hypothesis 2** (Causal Emergence). *The causal generative process of multi-value representation is composed of non-autonomous modules that inform or influence each other, meaning that the representation is causally entangled over all dimensions, that is, $E(\hat{x}_{s,m,n}) \doteq x_{s,m,n}$.*

Hypothesis 1 illustrates the natural linear order of indefinite data (e.g., $\{U_1, U_2, U_3, U_4\}$, where $U_1$ to $U_4$ respectively represent 4 utterances appearing in time-series, and $U_i \prec U_j$ indicates that $U_i$ precedes $U_j$ in time) belongs to the causal partial order. Consequently, the adjacency matrix of the natural linear order is a triangular matrix, which naturally corresponds to a DAG. Thus, there is no need for measures such as acyclic constraints (Zheng et al., 2018) to ensure causal identifiability.

Hypothesis 2 can provide insights into causal representation from the perspective of the law of large numbers. For statistical data, such as temperature, we need enough samples to grasp its characteristics (or distributions) in a particular environment. However, for non-statistical data, this is unnecessary. For instance, any sentence is enough to express its semantics, a single image can be read for its content. This hypothesis releases the limitation of insufficient samples, allowing us to achieve the causal consistency condition through strength sets.

## 2.2 WHY CAUSAL INCONSISTENCY ARISES?

Figure 1 visualizes evaluation results of causal consistency via tested 5 methods: PC (Kalisch & Bühlman, 2007), ACD (Löwe et al., 2022), DAG-GNN (Yu et al., 2019), CAE (Chen et al., 2023b), and biCD (Chen et al., 2023c). They represent prevalent methods in specific data forms, respectively. Two conclusions can be obtained from Figure 1: 1) The strongest causal inconsistency is found in indefinite data forms ($M > 1$, $D > 1$), while definite data ($M = 1$, $D = 1$) performs the weakest causal inconsistency. 2) When existing methods are applied to non-default data forms (hollow markers), their consistency performance is always inferior to the native methods for that data form.

Either $M > 1$ or $D > 1$ contributes to a rise in inconsistency. In general, when $M > 1$, the optimization for causal strength changes from $f$ to $\sum \alpha_m f_m$, which leads to a lower accuracy of causal structure than the ones of $M = 1$, due to the existence of Pareto Optimality (Censor, 1977). Meanwhile, $D > 1$ introduces deep representations, resulting in an inexact transforma-

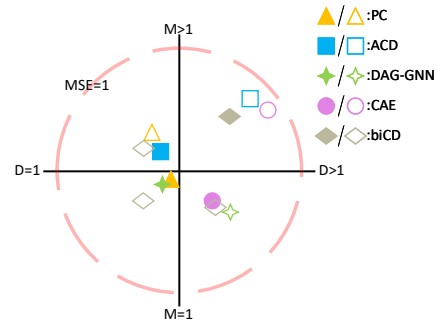

Figure 1: We compared the consistency of different methods in 3 data paradigms (if available). The consistency was represented by the MSE of the similarity matrices for structure and representation. The filled markers represent methods being in their default data forms, while the hollow markers signify that they are in extendable but non-default data forms.

tion. When both $M > 1$ and $D > 1$ are present, we assume $f_m$ can be decoupled from $\tilde{p}_\varphi$. The causal structure learning reads $\mathcal{G}_m = h(X, \varphi)$, and the causal representation learning is $\hat{X}_m = h(X, \varphi, \theta)$. The different learning processes with additional parameter $\theta$ intrinsically increase more inconsistency than other data paradigms. Details are elaborated on in Appendix A.3.

### 2.3 RELATED WORKS: CAUSAL METHODS ON DIFFERENT DATA PARADIGMS

**M=1 and D=1**: Methods applied to definite data (Janzing et al., 2012; Ramsey et al., 2017; Cai et al., 2020) can ensure consistency due to their effective utilization of statistical advantages.

**M>1 and D=1**: To address the challenge of multi skeletons, some methods try to learn an invariance, such as ACD (Löwe et al., 2022). This range of approaches (Huang et al., 2020a; Dhir & Lee, 2020; Huang et al., 2020b; 2019) uses neural networks to automate the learning the distribution of causal structure $\sum_{m=1}^{M} \log(P(X_m))$. Despite the promising performance of the whole structures, the accuracy of individual structures decreases thereby increasing inconsistency.

**M=1 and D>1**: Similarly, for the more favorable causal representation, assumption of the stationary structure becomes the most prevailing choice involved interpretability (Fan et al., 2022; Wu et al., 2022), relationship analysis (Chen et al., 2023b; Zhao et al., 2022a), and domain generalization (Lv et al., 2022; Jiang & Veitch, 2022). Among these methods, the critical cause of inconsistency stems from inaccuracies in recovering relations. Specifically, the complexity of the data makes it challenging to identify exact causal relationships so that none can achieve the same perfect level as that in definite data.

**M>1 and D>1**: The inconsistency is more a result of additional parameter in the learning processes of structures and representations. Especially, indefinite data is in its infancy, and current methods focus more on enhancing causal accuracy, with explorations into causal consistency yet to be conducted. For example, while DAG-GNN (Yu et al., 2019) can be applied to indefinite data, it does not yield satisfactory results compared to its default data form (M=1 and D=1). Methods such as CAE (Chen et al., 2023b) and biCD (Chen et al., 2023c), although they have improved causal accuracy, display significant causal inconsistency, particularly in advanced tasks with incomplete labels.

## 3 HOW TO CHECK CAUSAL CONSISTENCY

An observed variable $x_t$ of indefinite data does not satisfy $P(y|x_t = t_1)$, $P(y|x_t = t_2)$, ... without adequate samples $t = t_1$, $t = t_2$, ... so that the distribution is not clear for intervention. Hence, we define a general intervention, intending to bypass distribution assumptions to obtain interventions.

**Definition 2** (General Intervention). *General intervention is represented by the $do_g$ operator with the objective of setting the parent set of the observed variable to $\emptyset$.*

$$do_g(x_t) := Pa(x_t) = \emptyset \tag{1}$$

*where $Pa(x)$ represents a parent set of $x$.*

Benefitted from Hypothesis 2, effects of $do_g(x_t)$ are equivalent to effects of the set of perfect interventions: $\{do(x_t = t_1), do(x_t = t_2), \dots\}$. (For simplicity, unless specially stated, the term $do$ in the rest of this paper represent either $do$ or $do_g$.) Definition 2 introduces feasible intervention for indefinite data to allow us to draw inspiration from Definition 6 in Appendix A.1: the consistency of two causal model can be verified under any reasonable intervention. To ensure this idea, the causal models need to include intervention sets and strength sets.

**Definition 3** (Causal Model). *Let causal model $M_X = \langle S_X, I_X, F_X^{I_X} \rangle$, where $S_X$ represents an SCM for the model with the variable set $X = (x_i : i \in \natural_x)$, $\natural_x$ is the index of causal partial order over $X$, $I_X := (do(i,j) : i, j \in \natural_x, \preccurlyeq_{do})$ represents a set of all reasonable bi-variable perfect interventions satisfying partial order, $F_X^{I_X} := (f_X^{do(i,j)} : i, j \in \natural_x, \preccurlyeq_{do})$ represents the causal strength of set $X$ under corresponding interventions.*

Considering the presence of front-door paths, it is complicated to directly calculate the causal relationship between any two variables by intervention on just one variable. Consequently, we form the intervention set using bi-variable interventions. For example, in a binary definite data set, the intervention set could be $I_X = \{\emptyset, do(x_1 = 0, x_2 = 0), do(x_2 = 0, x_3 = 1), \dots\}$. In the indefinite data, the intervention set could be $I_X = \{\emptyset, do_g(x_1, x_2), do_g(x_2, x_3), \dots\}$. $\preccurlyeq_{do}$ represents that for any pair $< x_i, x_j >$ in $I_X$, where $x_I$ signifies the $x_i$ in the pair $< x_i, x_j >$ and all pairs previous to $< x_i, x_j >$, it is always in the causal partial order that $x_i$ does not follow $x_j$. The strength set $F_X^{I_X}$ would have the same partial order $\preccurlyeq_{do}$, representing any causal strength of model $M_X$ corresponding

to the perfect intervention of $I_X$. In definite data, according to the causal factorization mentioned in Appendix A.1, the strength set can be equated to the distribution set $\mathbb{P}_X^{I_X}$. Finally, we would like to introduce the causal consistency condition:

**Theorem 1** (Causal Consistency Condition (CCC)). *Let* $\mathcal{U}_X = (S_X, I_*, F_X^{I_*})$ *and* $\mathcal{V}_Y = (S_Y, I_*, F_Y^{I_*})$ *be two causal models. The intervention set* $I_*$ *denotes that there is an identity mapping between* $X$ *and* $Y$. *If any term* $f_{y_1,y_2}^{do(i,j)}$ *in* $F_Y^{I_*}$ *satisfies:*

$$f_{y_1,y_2}^{do(i,j)} = f_{x_1,x_2}^{do(i,j)} \tag{2}$$

*the* $\mathcal{U}_X$ *is consistent with* $\mathcal{V}_Y$ *(Proof is given in Appendix B).*

**Example 1.** *In indefinite data, we assume that the causal structure belongs to a definite causal model* $\mathcal{U}$, *and the causal representation belongs to an indefinite causal model* $\mathcal{V}$. *Since* $\mathcal{U}$ *and* $\mathcal{V}$ *have the same causal variables, there exists an order-preserving bijection* $\omega := I_X \Leftrightarrow I_Y$. *If* $\mathcal{U}$ *and* $\mathcal{V}$ *satisfy the CCC, let* $F_{\oslash} = (f_{a,b} : a, b \in \natural_x, (a \preccurlyeq_X i, b = i)\,or(a \preccurlyeq_X j, b = j), \preccurlyeq_{do})$ *w.r.t.* $do(i,j)$. *Any term* $f_{y_1,y_2}^{do_g(i,j)}$ *in* $F_Y^{I_Y}$ *satisfies* $f_{y_1,y_2}^{\omega(do(i,j))} = f_{x_1,x_2}^{do(i,j)}$ *and if the factorization of* $f_{y_1,y_2}^{\omega(do(i,j))}$ *includes* $f_{a,b} \in F_{\oslash}$, *it satisfies* $f_{y_1,y_2}^{\omega(do(i,j))} = f_{x_1,x_2}^{do(i,j)} = 0$. *The conclusion is also satisfied on* $I_Y$.

The SMS hypothesis proposed in Schölkopf et al. (2021) elucidated Example 1. Simply put, if the causal structure is robust, the interventionally-affected conditional probability can not influence the interventionally-unaffected conditional probability in causal factorization $P(x_1, x_2, \ldots, x_s) = \prod_{s=1}^{S} P(x_s | X_{Pa_{x_s}})$. On the contrary, the interventionally-unaffected conditional probability could not maintain stability if causal model is unrobust. Therefore, we not only require the strengths of the intervention nodes to be consistent ($f_{y_1,y_2}^{\omega(do(i,j))} = f_{x_1,x_2}^{do(i,j)} = 0$), but also that the strengths of nodes without induced paths to the intervention nodes stay consistent ($f_{y_1,y_2}^{\omega(do(i,j))} = f_{x_1,x_2}^{do(i,j)}! = 0$).

# 4 SSL FRAMEWORK

## 4.1 FORMULATION ARCHITECTURE

The causal structure and causal representation of each sample in the indefinite data can be viewed as belonging to two individual models. Specifically, the causal structure $\mathcal{G}_{s,m} = h(X_{s,m}, \varphi)$ can be seen as part of a definite data causal model $\mathcal{U} = (S_{X_{s,m}}, I_*, F_{X_{s,m}}^{I_*})$, while the causal representation $\hat{X}_{s,m} = h(X_{s,m}, \varphi, \theta)$ can be considered part of an indefinite data causal model $\mathcal{V} = (S_{\hat{X}_{s,m}}, I_*, F_{\hat{X}_{s,m}}^{I_*})$ ($\natural_{\hat{x}} = \natural_x$, $\preccurlyeq_{\hat{X}_{s,m}} = \preccurlyeq_{X_{s,m}}$). Therefore, there are two causal models ($\mathcal{U}$ and $\mathcal{V}$) correspondingly with two causal structures ($\mathcal{G}_{s,m}$ and $\hat{\mathcal{G}}_{s,m}$) and two causal representation ($X_{s,m}$ and $\hat{X}_{s,m}$). The aim of SSL framework is to establish equivalent strength set $F_{X_{s,m}}^{I_*} = F_{\hat{X}_{s,m}}^{I_*}$, thereby achieving causal consistency $\mathcal{U} = \mathcal{V}$. We elucidate the roles of "view", "augment", and "philosophy" within our framework as follows.

**View**:We define an intervention ($do(i,j) \in I_*$) as a view. For example, $do(x_1, x_2)$, $do(x_2, x_3)$, and $do(x_1, x_3)$ could be 3 individual views of $\mathcal{U}$; $do_g(x_1, x_2)$, $do_g(x_2, x_3)$, and $do_g(x_1, x_3)$ could be ones of $\mathcal{V}$.

**Augment**:We define the specific measures for obtaining the $F_{\hat{X}_{s,m}}^{I_*}$ under intervention as the augments. e.g., $f_{\hat{X}}^{do_g(i,j)} = augment_{do_g(i,j)}(\hat{X})$. $F_{X_{s,m}}^{I_*}$ can be directly obtained via checking $\mathcal{G}_{s,m}$ or $S_{X_{s,m}}$ because it belongs to the definite-data causal model.

**Philosophy**:We define the causal consistency between $\mathcal{U}$ and $\mathcal{V}$ as the philosophy. e.g., $\mathcal{U}$ and $\mathcal{V}$ should satisfy the CCC (Theorem 1): for any view, $f_{\hat{X}}^{do_g(i,j)} = f_X^{do(i,j)}$.

Overall, as illustrated in Figure 2. The views of causal structure and causal representation correspond with each other. The "augment" process derives strength sets separately under these views, which are then evaluated for consistency according to the philosophy of causal consistency. Within this, both the "Augment" and "Consistency check" modules in Figure 2 depend on specific implementation. The "Augment" should not introduce any new parameters, otherwise the optimization would be:

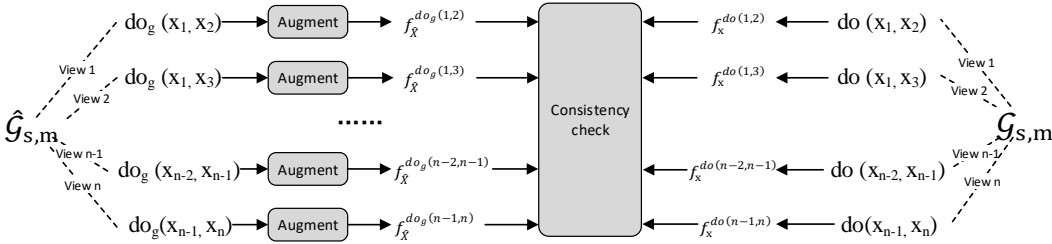

Figure 2: The SSL framework for causal consistency. The grey rectangular boxes represent modules that require specific implementation. From left to center, the process describes how causal representations are transformed into causal strengths. From right to center, the process illustrates how causal structures are converted into strengths.

Table 1: Number of the samples in *Causalogue* Dataset

| Versions | Structure Types | | | | | | | | | | |
|---|---|---|---|---|---|---|---|---|---|---|---|
| | Chain_I | Chain_II | Chain_III | Chain_IV | Fork_I | Fork_II | Fork_III | Fork_IV | Hybrid_I | Hybrid_II | Total |
| Small | 276 | 84 | 141 | 44 | 257 | 237 | 251 | 67 | 185 | 77 | 1638 |
| Large | 0 | 524 | 508 | 513 | 1215 | 645 | 501 | 372 | 499 | 635 | 5412 |

$\min_{\theta, \delta_1, \ldots, \delta_K} \sum_{s=1}^{S} \sum_{m=1}^{M} \sum_{k=1}^{K} \mathcal{L}_k(\hat{\mathcal{G}}_{s,m}, \delta_k, \theta)$. The "consistency check" module shares parameters and does not introduce intervention-unique parameters. Therefore, the optimization of our entire SSL process is written as:

$$\min_{\theta} \sum_{s=1}^{S} \sum_{m=1}^{M} \mathcal{L}_k(\hat{\mathcal{G}}_{s,m}, \theta) \tag{3}$$

### 4.2 TWO EXAMPLES FOR IMPLEMENTATION

We provide two implementation examples. The first one is on a supervised specialized model (**SSM**) generating high-level causal representations and structures. The second is implemented on an unsupervised large language model (**LLM**), which can be used to directly infer causal relationships between utterances (as dialogues are typically indefinite data). The consistency check modules in both examples are accomplished through similarity matrices, yet the augment modules are completely different: the first example computes strength by modifying adjacency matrices, while the second example offers two approaches: prompts and pre-trained models. Detailed implementation specifics are thoroughly described in Appendix C.

## 5 NEW SIMULATION DATASET-*Causalogue*

Existing indefinite datasets suffer from issues including incomplete labeling and insufficient samples. These numerous entangled problems make it challenging to achieve pure evaluation for the inconsistency. Additionally, the challenge of manual annotation is considerable, as the presence of numerous ambiguous samples could make classification boundaries unstable. Fortunately, the powerful human-computer conversation abilities of LLMs, such as GPT-4, have made automated annotation possible. Thus, to provide a high-quantity dataset for checking consistency, we have made an endeavor to generate controlled, causal dialogues via GPT-4 ending up with a new dataset, *causalogue*. This is the first dialogue dataset that includes comprehensive causal relationship labels for indefinite data. Besides causality-related tasks, the dataset is available for all tasks related to dialogue relationships (e.g., dialogue generation, relation extraction, and text classification).

The dataset incorporates 10 types of causal structures ($M = 10$), each with several samples (Detailed number are presented in Table 1, "Small" signifies samples that have been manually checked as correctly labeled, while "large" refers to all samples generated by GPT-4 without manual verification).

Table 2: Summarization of datasets and baselines

| Ours | Tasks | Datasets | Baselines | Metrics |
|------|-------|----------|-----------|---------|
| $\text{Ours}_{SSM}$ | CD | *Causalogue* | ACD, DAG-GNN, ACCD, biCD, DisC, DIR | AUROC, HD, F1, MSE |
| | ECPE | RECCON | ACCD, biCD, EDKA-GM, seF | F1 |
| | ERC | MELD, EmoryNLP, DD, IEM | ACCD, biCD, DAG-ERC, DualGAT, MultiEMO | F1 |
| | TAS | GTEA, 50salads, Breakfast | MS-TCN++, ASRF, CETNet, C2F | acc, Edit, F1@k,C-Dis |
| $\text{Ours}_{LLM}$ | CD | *Causalogue*, RECCON | Zero-shot, Zero-shot-Cot, Auto-Cot | F1 |

All samples consist of 4 causal variables. In each sample, binary causal relationships have been labeled between any two utterances. A detailed exposition of the dataset's attributes and creation process can be found in Appendix D.

# 6 EXPERIMENTS

## 6.1 DATASETS, BASELINES, AND METRICS

Including the Causal Discovery (CD) task on *Causalogue* dataset, we also evaluate our method on real-world datasets (RECCON (Poria et al., 2021), MELD (Poria et al., 2019), EmoryNLP (Zahiri & Choi, 2018), DD (Li et al., 2017), IEM (Busso et al., 2008), GTEA (Fathi et al., 2011), 50salads (Stein & McKenna, 2013), and Breakfast (Kuehne et al., 2014).) spanning three downstream tasks (Emotion-cause Pair Extraction (ECPE) task, Emotion Recognition in Conversation (ERC) task, and Temporal Action Segmentation (TAS) task) involving both text and video.

The experiments also incorporate a variety of baselines. For the supervised specialized models (SSMs), it encompasses causal deep models such as ACD (Löwe et al., 2022), DAG-GNN (Yu et al., 2019), ACCD (Chen et al., 2023b), biCD (Chen et al., 2023c), and intervention deep models like DisC (Fan et al., 2022), DIR (Wu et al., 2022), and our example ($\text{Ours}_{SSM}$ in Appendix C.1). Moreover, we evaluate the downstream tasks with additional SOTA work pertinent to each task, such as EDKA-GM (Li et al., 2023a), seF (Li et al., 2023b) for ECPE task, DAG-ERC (Shen et al., 2021), DualGAT (Zhang et al., 2023a), MultiEMO (Shi & Huang, 2023) for ERC task, and MS-TCN++ (Li et al., 2020), ASRF (Ishikawa et al., 2021), CETNet (Wang et al., 2023), and C2F (Singhania et al., 2021) for TAS task. In terms of LLMs, we compared prompt-based baselines: Zero-Shot (Kojima et al., 2022), Zero-Shot-Cot (Kojima et al., 2022), Auto-Cot (Zhang et al., 2022), and our example($\text{Ours}_{LLM}$ in Appendix C.2). on public GPT-4 of the gpt-4-32k-0314 version.

For the CD task, we employed an array of metrics: Area Under the Receiver Operating Characteristic curve (AUROC), F1 score, Hamming Distance (HD), and Mean Squared Error (MSE) to comprehensively evaluate both the precision and consistency of causality. For different downstream tasks, we utilized their prevalent metrics for evaluations.

The criss-cross relationships between these datasets, tasks, metrics, and baselines have been summarized in Table 2 and the details of them are shown in Appendix E).

## 6.2 IMPLEMENTATION DETAILS

For the SSM, different pre-training models and implementation parameters were adopted for different downstream tasks. Detailed descriptions of these variants are provided in Appendix E.4. As for LLM, we solely implemented a prompt instruction method without adjusting the model parameters. The models include gpt-4-32k-0314.

## 6.3 SIMULATED DATASET

### 6.3.1 RESULTS OF THE SSMs

We conducted experiments for causal accuracy and consistency of the SSMs on the *Causalogue* dataset. Causal accuracy was evaluated through the performance of causal graphs and causal representations. Causal consistency, was assessed by measuring the distance within the similarity matrix between the graphs and representations. As demonstrated by Table 3, our method significantly improved causal consistency, which correspondingly led to an enhancement of causal accuracy. This also elaborates

Table 3: Results of SSMs on *Causalogue* Dataset. 95% confidence interval shown. All evaluation metrics (except HD) were normalized to the range $[0, 1]$. A value closer to 1 indicates better performance. HD (Hamming Distance) measure the corrected edges from results to labels.

| Methods | Causal Structure | | Causal Representation | | Causal Consistency | |
|---|---|---|---|---|---|---|
| | AUROC | HD | AUROC | F1 | AUROC | 1-MSE |
| ACD | $0.84_{\pm 0.02}$ | $0.89_{\pm 0.21}$ | $0.85_{\pm 0.02}$ | $0.88_{\pm 0.01}$ | $0.51_{\pm 0.01}$ | $0.49_{\pm 0.01}$ |
| DAG-GNN | $0.56_{\pm 0.04}$ | $1.51_{\pm 0.34}$ | $0.90_{\pm 0.01}$ | $0.88_{\pm 0.02}$ | $0.50_{\pm 0.01}$ | $0.49_{\pm 0.02}$ |
| DAG-DisC | $0.68_{\pm 0.27}$ | $1.40_{\pm 0.42}$ | $0.88_{\pm 0.02}$ | $0.87_{\pm 0.02}$ | $0.52_{\pm 0.00}$ | $0.50_{\pm 0.01}$ |
| DAG-DIR | $0.67_{\pm 0.38}$ | $1.36_{\pm 0.36}$ | $0.89_{\pm 0.03}$ | $0.86_{\pm 0.03}$ | $0.51_{\pm 0.01}$ | $0.50_{\pm 0.01}$ |
| ACCD | $0.79_{\pm 0.11}$ | $1.02_{\pm 0.15}$ | $0.93_{\pm 0.01}$ | $0.92_{\pm 0.03}$ | $0.60_{\pm 0.05}$ | $0.59_{\pm 0.11}$ |
| biCD | $0.91_{\pm 0.04}$ | $0.56_{\pm 0.10}$ | $0.86_{\pm 0.03}$ | $0.89_{\pm 0.02}$ | $0.64_{\pm 0.04}$ | $0.59_{\pm 0.07}$ |
| $\text{Ours}_{SSM}$ | $\mathbf{0.94}_{\pm 0.01}$ | $\mathbf{0.29}_{\pm 0.05}$ | $\mathbf{0.94}_{\pm 0.01}$ | $\mathbf{0.95}_{\pm 0.01}$ | $\mathbf{0.95}_{\pm 0.01}$ | $\mathbf{0.92}_{\pm 0.01}$ |

the fact that, until now, the causal consistency in indefinite data has been often overlooked though it is a crucial problem.

Our findings also indicates some additional conclusions. As shown between three 'DAG'-related baselines, the intervention methods proposed by DisC and DIR could enhance the causal graph identification capability. This improvement is attributed to that interventions can underlyingly adapt models to cross i.i.d. environment. However, their interventions introduced bias that lies in forming negative samples by combining the causal pattern with the background from other samples in the batch, when CCC sets it as $\emptyset$. Their intervention concepts do enhance the model's discriminative capacity for causal patterns and shortcuts, but the sparsity of indefinite data samples has been introduced as bias into contrastive learning. In addition, ACD and biCD are methods specifically targeted at multi-value and multi-skeleton data, respectively. Therefore, they have been particularly emphasized in our experimental results.

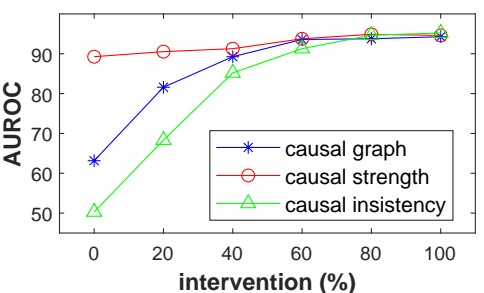

Figure 3: Performance of the $\text{Ours}_{SSM}$ under different sizes of intervention. "Intervention 20%" refers to an intervention set composed by randomly selecting 20% $do$ operators from the maximum intervention set.

To evaluate the volume of the interventions to results, we tested the performance under conditions ranging from an empty intervention set (no interventions carried out) to the maximum intervention set (all interventions carried out). Figure 3 demonstrates that interventions can significantly enhance causal consistency, thereby improving causal accuracy. Moreover, the size of the complete intervention set is close to 70% of the maximum intervention set, which is also reflected in the Figure 3 as a notable stability after 70%.

Lastly, we conducted an ablation study to determine the contributions of each mechanism. Table 4 demonstrates that the specific implementation of the module contributes little, but the causal mechanism (-adj) and causal identifiability (-mask) are the primary contributors to the causal discovery in indefinite data. However, the causal representation benefits more from fitting capacity, hence the decreasing induced by the causal mechanism is not as evident as the other two measures.

Table 4: Ablation Results of AUROC on three measures. '-cos_sim': replacing cosine similarity with MSE, '-adj' Removing matrix $A$, '-mask': replacing Hypothesis 1 with no acyclic constraints.

| Model | Structure | Representation | Consistency |
|---|---|---|---|
| -cos_sim | ↓ 0.02 | ↓ 0.01 | ↓ 0.00 |
| -adj | ↓ 0.34 | ↓ 0.12 | ↓ 0.43 |
| -mask | ↓ 0.46 | ↓ 0.11 | ↓ 0.47 |

### 6.3.2 RESULTS OF THE LLMS

$\text{Our}_{LLM}$, was tested on the *Causalogue* and RECCON datasets, with a simple set of experiments detailed in Appendix F. Specifically, we assessed the the upper bound of accuracy of varying methods to calculate $Sim^r$, and recorded the performance of our model as it approaches this bound. Finally, we illustrated specific question-answer content through a case analysis.

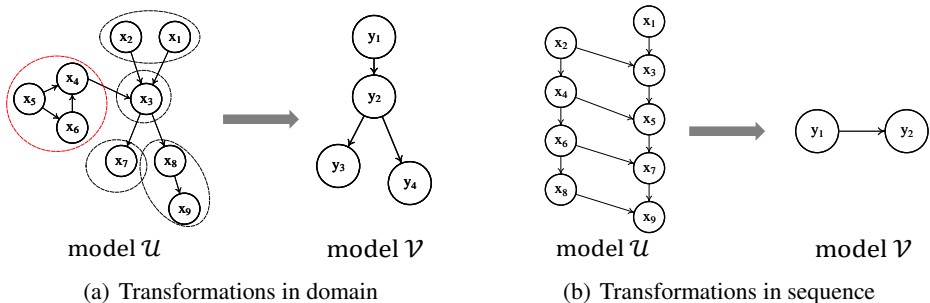

(a) Transformations in domain        (b) Transformations in sequence

Figure 4: Two potential insights of general causal consistency condition. Black elliptical frames represents the clustering of similar nodes, and red one represents marginalization of irrelevant nodes.

## 6.4 Real-world Datasets

We analyzed performance on three total downstream tasks, of which two are text modal: ECPE and ERC, and one is video modal: TAS. Comparative results against corresponding SOTA baselines, along with visualization cases, can be found in Appendix G. The results collectively suggest that $\text{Ours}_{SSM}$ performs well not only under experiment-environment causal models, but also enables more appropriate inference under high-level causal models.

## 7 Discussion: More Insights and Contributions

Focusing on addressing the SSL framework for causal consistency of indefinite data, this paper contributes to general intervention measures for distribution-unknown data, novel dataset, and reasoning on LLMs. However, considering the evolving progression of causal abstraction, our study only represents a beginning. The causal consistency condition (CCC) can only be applied to causal models with identical variable sets, whereas causal abstraction in definite data can already process models featuring two distinct variable sets— marking a significant discrepancy in contribution. Expanding causal abstraction to indefinite data (turning CCC into general CCC) could accurately search simplified models for intractable and complex models, thereby providing insights for many fields of deep learning.

Figure 4 describes two applications of general CCC-in domains and sequences, illustrating a new function of transformations from complex to simple models which are not confined to mere consistency checks. Specifically, transformations in domains primarily include clustering and marginalization in Figure 4 a: $\tau : (x_1, x_2) \rightarrow (y_1), (x_3) \rightarrow (y_2), (x_7) \rightarrow (y_3), (x_8, x_9) \rightarrow (y_4)$. This could solidify theoretical backing for established methods within various fields if a $\omega$ mapping exists on the intervention sets to make the strength sets equal. One notable application is interpretability of graph neural networks (Fan et al., 2022; Wu et al., 2022), where the goal is distinguishing between causal and other patterns amidst numerous nodes. Actually, we have explored the mapping from micro to macro model in the TAS task: a frame corresponding to a variable in the micro model $\mathcal{U}$, while a segment being a variable in the macro model $\mathcal{V}$, striving to construct equivalent mappings from frame to segment. The concept that the simplified model $\mathcal{V}$ can be viewed as the emergence of macro relationships in complex system $\mathcal{U}$, holding potential in researching areas such as meta-learning and domain generalization. Transformations in sequence, as shown in Figure 4 b are closely tied to temporal causal discovery, expressing dynamic processes through stable behaviors—a crucial aspect of sequence-to-sequence models and time-series forecasting.

However, there are significant challenges in implementing general CCC both theoretically and practically. To accomplish the case of "identity mapping" in causal abstraction on indefinite data, we introduced innovative intervention measures, frameworks, and validation methods. The general solution mentioned above requires much more theoretical research and experimental evaluation than this paper offers, and we eagerly anticipate sharing these findings in the future. In summary, preserving causal consistency in indefinite data marks an inventive and promising initiation in aligning causal theory with the deep learning.

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

## A  SUPPLEMENTARY EXPLANATIONS FOR CAUSAL DATA

### A.1  PRELIMINARIES

The joint distribution of samples within the same structure can be represented by any factorization:

$$P(x_1, x_2, \ldots, x_s) = \prod_{s=1}^{S} P(x_s | X_{others}) \tag{4}$$

and it can always be consistent with the probability distribution of a certain graph. For instance, in statistical models, for any variable $x_j \in X_{others}$, there exists an undirected edge between $x_j$ and $x_i$, representing the correlation between $x_j$ and $x_i$. Despite the Markov property, the direction can't be directly identified by the conditional probability for an undirected edge. However, the causal model can identify the causal direction between two related variables $x_j$ and $x_i$ by intervention (for example, $x_j \to x_i$, $x_i \to x_j$, $x_j \to L \to x_i$, $x_j \leftarrow L \to x_i$ and so on).

**Definition 4** (Intervention). *Interventions are typically represented by the do operator, with the objective of setting the probability of an observed variable equaling a particular state to 1.*

$$do(x_t = t) := P(x_t = t) = 1 \tag{5}$$

*where $t$ is one of the state which probably exist in the original distribution of variable $x_t$.*

When the interventions are integrated with factorization (Equation 4), the factorization satisfies causal factorization, i.e., the factorization can be converted into its corresponding Structural Causal Model (SCM).

**Definition 5** (Structural Causal Model). *An SCM is a 4-tuple $\langle X, \mathcal{F}, U, \mathbb{P} \rangle$, where $X$ is the entire set of observed variables $X = \{x_i\}_{i=1}^{S}$. Structural equations $\mathcal{F} = \{f_i\}_{i=1}^{S}$ are functions that determine causal representation $\hat{X}$ with $\hat{x}_i = f_i(Pa(x_i), u_i)$, where $Pa(x_i) \subseteq X$ represents the parent set of $X$, $u_i \in U$ represents the i.i.d. noise term. $\mathbb{P}(X)$ is a distribution set over $X$.*

Exact transformation (Rubenstein et al., 2017) or causal abstraction (Beckers & Halpern, 2019) is a method of judging causal consistency based on interventions and SCMs. The $\tau$ transformation becomes vital for making two causal models equivalent.

**Definition 6** ($\tau$-transformation). *Let $I_L$ to be a set of interventions on micro model $SCM_M = \langle X_M, \mathcal{F}_M, U_M, \mathbb{P}_M \rangle$. Similarly, let $I_N$ be interventions on macro model $SCM_N = \langle X_N, \mathcal{F}_N, U_N, \mathbb{P}_N \rangle$. Let $\tau$ be a partial transformation function $\tau : \mathbb{P}_M(X_M) \to \mathbb{P}_N(X_N)$. Let $\omega : I_M \to I_N$ be*

$$\tau(\mathbb{P}_M(X_M)) \to \mathbb{P}_N(X_N) = \omega(I_M) \to I_N \tag{6}$$

### A.2  EXAMPLES AND CHARACTERISTICS OF CAUSAL DATA

**Example 2** (Definite Data). *Arrhythmia Dataset (Guvenir et al., 1997) is a case record dataset from patients with arrhythmias, including 452 samples, and each sample consists of 279 single-value variables (e.g., age, weight, heart rate, etc.). All samples contribute a common causal graph with 279 nodes, where the edge value indicates some causal relationship, such as the causal strength of how age affects heart rate.*

**Example 3** (Semi-definite Data (Multi-skeleton and Single-value)). *The Netsim dataset (Smith et al., 2011) is a simulated fMRI dataset. Because different activities in brain regions over time imply different categories, a set of records of one patient corresponds to one causal sturcture. This dataset includes 50 sturctures and each sturcture consists of 15 single-value variables that measure the signal strength of 15 brain regions.*

**Example 4** (Semi-definite Data (Single-skeleton and Multi-value)). *CMNIST-75sp (Fan et al., 2022) is a graph classification dataset with controllable bias degrees. In this dataset, all researchers concentrate on one causal graph including 4 variables: causal variable $C$, background variable $B$, observed graph $G$ and label $Y$. $C$ is a part of the MNIST image including multi value of a group of pixels.*

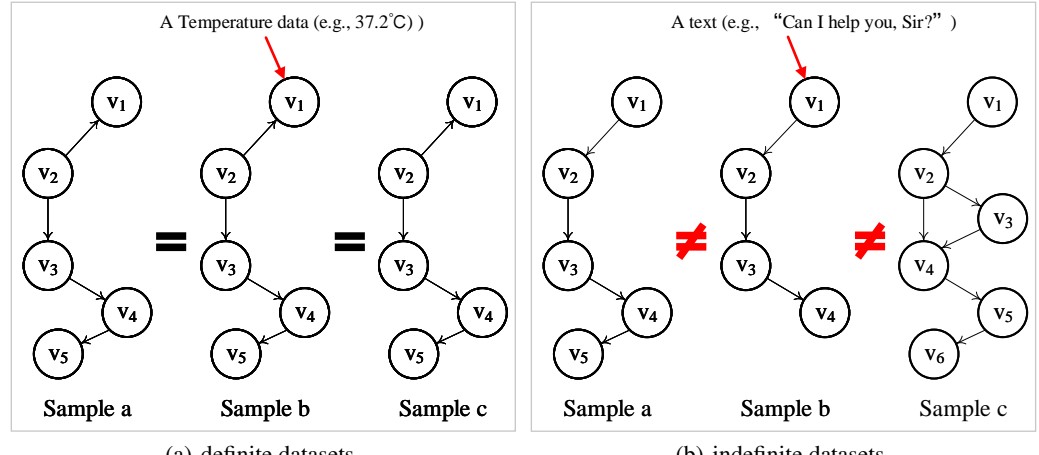

(a) definite datasets        (b) indefinite datasets

Figure 5: Differences between definite datasets (where $M = 1$ and $D = 1$) and indefinite datasets (where $M > 1$ and $D > 1$). In definite datasets, each sample corresponds to an identical causal structure, implying a single-skeleton trait as the entire dataset involves only a single causal structure. In contrast, indefinite datasets do not possess one causal structure for all samples. For instance, there might be varying numbers of causal variables in samples a, b, and c; the relationship $v_2 \rightarrow v_4$ may be absent in sample b but present in sample c. Furthermore, the causal variables in definite and indefinite datasets also differ. For example, in definite datasets, the causal variable $v_1$ might represent body temperature, with a causal representation of 37.3 in sample a, 37.1 in sample b and 36.8 in sample c, while $v_2$ might symbolize blood pressure, with a causal representation of 118, 127, and 135 separately; they are both single-value data. However, in indefinite data, within a dialogue dataset, the causal variable $v_1$ might be an utterance ("Can I help you, Sir?" in sample a, "Nice to meet you !" in sample b, and "What's the matter with you" in sample c) with its causal representation being a 768-dimension word embedding in deep model, and $v_2$ might be a responding utterance to $v_1$. In a video dataset, $v_1$ might denote a segment representing a particular action or event, with its causal representation as the corresponding optical flow, and $v_2$ might be another segment triggered by $v_1$.

**Example 5** (Indefinite Data). *IEM Dataset (Busso et al., 2008) is a conversation record dataset with each sample including a dialogue between two speakers. All 100 samples are assigned into 26 graphs (i.e., 26 skeletons) based on the speaker identifies and turns and each sample consists of 5-24 variables where each variable is an utterance represented by embeddings $\in \mathbb{R}^{1 \times 768}$ or $\mathbb{R}^{1 \times 1024}$ in prevalent pretrained language models.*

We aim to illustrate the relationships among three data paradigms through Examples 2,3,4,5 and Figure 5, focusing particularly on the number of skeletons (single or multi-skeleton) and the dimension of causal representations (single or multi-value).

**single or multi-skeleton**: Compared to single-skeleton data (M = 1), multi-skeleton data (M > 1) lacks discrimination about which samples belong to the same causal structure. Therefore, it requires algorithms capable of distinguishing between different causal structures or clustering similar samples. Simultaneously, multi-skeleton data often have trouble in low sample utilization since samples from other skeletons contribute nothing when identifying a specific causal structure. Consequently, the pathways focusing on single-skeleton and multi-skeleton data are different.

**single or multi-value**: multi-value data (D > 1) often facilitate the quantification by deep representation, such as text → embeddings, image → matrices, audio → spectrum map, and video → optical flow, as exemplified in our Figure 5. Compared to single-value data (D = 1), it involves more complex environments. The statistical advantages of single-value data are more significant, such as computing independence between two single-value variables. On the contrary, determining such "independence" among multi-value representations is challenging, often approximated through algorithms like cosine similarity. In Structural Causal Models (SCMs), one can assume that the noise of single-value data follows a specific distribution, but in multi-value data, the noise items

629 are multi-value and interdependent among dimensions, causing many traditional causal discovery
630 methods to make no efforts with multi-value data.

### A.3 THE DISTINCTIONS AMONG THREE DATA PARADIGMS

632 Specifically, we employ the theory illustrated in Schölkopf et al. (2021) to explicate why the skeleton
633 (**M**) and variable dimension (**D**) are pivotal in capturing differences in causal discovery algorithms.
634 Accroding to the assumption in Schölkopf et al. (2021), the domain of causal variables $\mathcal{X}$ is projected
635 onto the domain of causal representations $\hat{\mathcal{X}}$ via the encoder $p_\varphi$ and decoder $q_\theta$, showcasing the
636 causal mechanism in structural equations:

$$\hat{x}_i = f_i(Pa_i, U_i) \tag{7}$$

637 where $Pa_i$ represent the parent node set of $x_i$. For instance, $p_\varphi : U = (1 - A)X$ and $q_\theta : \hat{X} =$
638 $(1 - A)^{-1}U$. Without prior knowledge, there exist two pathways to recover the causal model: 1)
639 Given a fixed causal structure and known causal representation, the causal strength can be estimated
640 by the statistical strength observable in the samples. 2) If encoder and decoder are feasible, optimal
641 solutions of the causal model can be achieved by minimizing the reconstruction loss $p_\varphi \circ f \circ q_\theta$. Here
642 we would like to delimit the solvability of this process for different combinations via M=1, M>1,
643 D=1, and D>1.

644 **For a single-skeleton model (M=1)**: When the causal structure is fixed, causal strengths $f$ can be
645 calculated. If the causal representation is single-value (D=1), the causal structure can be determined
646 without the encoder $p_\varphi$ or decoder $q_\theta$. The reconstruction loss in this case is $f$. However, for
647 multi-value data (D> 1), in the reconstruction loss function $p_\varphi \circ f \circ q_\theta$, $f$ represents the being
648 determined part.

649 **For a multi-skeleton model (M> 1)**: The multi-skeleton data induce uncertainty in causal structures,
650 unclear of which samples correspond to the same causal structure and therefore making causal
651 strengths $f$ unsolvable directly. However, under single-value (D=1) condition without generated
652 representation, the precision of clustering is guaranteed. We can approach by first clustering the
653 samples, and then separate the problem to several tasks of definite data problem-solving (M=1,
654 D=1). In this regard, reconstruction loss amounts to $\{f_m\}_{m=1}^M$, representing the set containing
655 each sub-task's $f_m$. Reconstruction loss can be regarded as a multi-task optimization problem,
656 $\alpha_1 f_1 + \alpha_2 f_2 + \cdots + \alpha_M f_M$, where $\alpha_m$ is the weights of the sample quantity per structure. The
657 worst-case scenario arises with multi-value data (D>1), only able to attain an approximate encoder
658 $\tilde{p}_\varphi = p_\varphi \circ f_m$, which results in a final reconstruction loss of $\tilde{p}_\varphi \circ q_\theta$. Causal strength $f_m$ comprises
659 an unassigned part.

660 In summary, for definite data ($M = 1$, $D = 1$), it suffices to identify the causal strength between
661 any two causal variables under a certain causal structure. Semi-definite data addresses the problem
662 of discriminating multi-skeleton structures and encoding multi-value variables separately. As for
663 indefinite data, in the absence of additional assumptions, causal discovery in such datasets presents
664 an ill-posed problem, given it requires both variable encoding and resolving structure discernibility.

## B  PROOF OF THEOREM 1

666 Existing work (Hu & Tian, 2022) proposed that two causal models satisfying causal abstraction are
667 consistent. Accordingly, we employ causal abstraction as a mediator to prove Theorem 1. In other
668 words, it suffices to verify that the concepts of equivalent distribution sets and equivalent strength
669 sets are conditionally transformable.

670 First, we would like to prove that equivalent distribution sets $\rightarrow$ equivalent strength sets.

671 According to the causal factorization, $P(x_s|X_{others})$ satisifies $x_s = f_s(Pa(x_s, u_s))$. When we
672 assume two causal model within two variables: $\mathcal{U} : x_i \rightarrow x_j$ and $\mathcal{V} : y_i \rightarrow y_j$, we would like to
673 adopt SCM to represent the equivalent distribution $P_{x_j}(do(x_i)) = P_{y_j}(\omega(do(x_i)))$:

$$f_{i,j}(u_{x_i}) + u_{x_j} = g_{i,j}(u_{y_i}) + u_{y_j} \tag{8}$$

Because of $P_{x_j}(do(x_j)) = P_{y_j}(\omega(do(x_j)))$ and $P_{x_i}(do(x_i)) = P_{y_i}(\omega(do(x_i)))$, Equation 8 can be writen as:

$$f_{i,j} = g_{i,j} \tag{9}$$

According to the causal partial order, $P_{x_j}(do(x_j)) \preccurlyeq P_{x_j}(do(x_j, x_k)), (x_j \preccurlyeq x_k)$, hence:

$$f_{j,k} = g_{j,k} \tag{10}$$

When we convert any causal factorization into a chain of ancestral relationships through additive noise formulas, it is possible to find a corresponding $g_{i,j}$ that equals $f_{i,j}$ for any step in the causal chain. Finally, we can infer that if the distribution sets $P_X = P_Y$, then the strength sets $F_X = F_Y$. Conversely, it can also be proven that if the strength sets $F_X = F_Y$, then the distribution sets $P_X = P_Y$.

## C    DETAILS OF TWO EXAMPLES FOR IMPLEMENTATION

### C.1    EXAMPLES IN SUPERVISED SPECIALIZED MODEL (SSM)

#### C.1.1    PREVALENT PROBABILISTIC MODEL

Many variational models for causal discovery, including linear SEM variational model (Yu et al., 2019), autoregressive (Wang et al., 2020) and recently substitude of noise (Chen et al., 2023b), can be encapsulated by a probabilistic framework:

1. Construct a Linear Structural Equation Model (SEM) to displace SCM. Specifically, let A $\in \mathbb{R}^{N \times N}$ be the adjacency matrix, and $N$ stands for the number of variables. $X \in \mathbb{R}^{N \times D}$ is a sample of $N$ variables.
$$X = AX + E \tag{11}$$
where $E \in \mathbb{R}^{N \times D}$ is the matrix of independent noise $\epsilon_{x_n}$, $A$ represents the causal strength from all variables to one observed variable.

2. Build a pair of Autoregression SEMs:
$$E = (I - A)X \tag{12}$$

$$X = (I - A)^{-1}E \tag{13}$$

   Equation 13 describes a general form as a decoder of a generation model that takes noise $E$ as input and returns $X$ as results and Equation 12 describes the corresponding encoder.

3. Considering a specification of noise ($E$) distribution sampling $\{X_s\}_{s=1}^{S}$ in definite data, Equation 13 can be written by a maximization of log-evidence:

$$\frac{1}{S}\sum_{s=1}^{S} \log p(X_s) = \frac{1}{S}\sum_{s=1}^{S} \log \int p(X_s|E)p(E)dE \tag{14}$$

   Continuing the theory of variational Bayes, we regard $E$ as the latent variable in variational autoencoder (VAE) (Kingma & Welling, 2022) and use variational posterior $q(E|X)$ to approximate the intractable posterior $p(E|X)$, thus the evidence lower bound (ELBO) reads:

$$\mathcal{L}_{ELBO}^{s} = -KL(q(E|X_s)||p(E)) + E_{q(E|X_s)}[\log p(X_s|E)] \tag{15}$$

### C.1.2    OUR SUPERVISED IMPLEMENTATION

Taking an example from the variational probabilistic framework mentioned in Appendix C.1.1, which has become a popular choice, we simplify consider Equation 12 as the encoder $p_\varphi$ and Equation 13 as the decoder $q_\theta$. The model $\mathcal{U}$ is generated by encoder: $\mathcal{G}_{s,m} = h(X_{s,m}, \varphi)$ and the model $\mathcal{V}$ is generated by encoder and decoder: $\hat{X}_{s,m} = h(X_{s,m}, \varphi, \theta)$. The adjacency matrix $A$ represents the influence between the observed variables. For example, $A_{i,j} \in [0, 1]$ describes the strength of

how the variable $x_j$ influences the variable $x_i$. The 'augment' measure corresponding to $do_g(i,j)$ is defined as follows:

$$A_{m,n} = \begin{cases} 0, & (m = i \ or \ m = j) \\ A_{m,n}, & else \end{cases} \tag{16}$$

The $i$-th and $j$-th row represent the influence of all parent nodes on variables $i$ and $j$, respectively. After the 'augment' measures, the corresponding $i$-th and $j$-th rows in the adjacency matrix $W = (1-A)^{-1}$ of the decoder only contain the two non-zero terms, $u_i$ and $u_j$. For the consistency check, we adopt an easily computable similarity matrix $Sim \in \mathbb{R}^{N \times N}$, where $Sim_{i,j}$ represents the similarity between variables $x_i$ and $x_j$. The similarity matrices of models $\mathcal{U}$ and $\mathcal{V}$ are respectively generated from distinct resources: model $\mathcal{U}$ and its similarity matrix are computed from causal representation, while model $\mathcal{V}$ and its similarity matrix are derived from the causal structure.

The similarity matrix of $\mathcal{U}$ is computed via cosine similarity:

$$Sim_{m,n}^r = cossim(\hat{x}_m^{do_g(i,j)}, \hat{x}_n^{do_g(i,j)}) * Mask \tag{17}$$

where $\hat{x}_m^{do_g(i,j)}$ and $\hat{x}_n^{do_g(i,j)}$ are the causal representation under the view $do_g(i,j)$, $Mask$ stands for a lower triangular matrix. $Mask_{i,j} = 0$ when $j > i$ and $Mask_{i,j} = 1$ when $j \leqslant i$.

The similarity matrix of $\mathcal{V}$, its can be obtained directly from the causal strength matrix:

$$Sim_{m,n}^s = \begin{cases} 0, & (n \leqslant i, m = i \ or \ n \leqslant j, m = j) \\ P(x_m^{do_g(i,j)} | x_n^{do_g(i,j)}), & else \end{cases} \tag{18}$$

Finally, the Consistency check module needs to measure these two similarity matrix within specifically MSE loss function we adopted:

$$Loss = MSE(Sim_{m,n}^r, Sim_{m,n}^s) \tag{19}$$

## C.2 Examples for Large Language Model (LLM)

Large language models (LLMs) demonstrate superior performance across a variety of text tasks, particularly showing "natural" level in human-machine conversations based on instrcutions. Considering dialogue as a typical data type of indefinite data, we have implemented an example where LLMs can identify causal relationships between utterances through prompt instructions.

Specifically, we treat each dialogue as a sample, where an utterance is regarded as a causal variable. That is, for a dialogue $D = \{Utt_1, Utt_2, \ldots, Utt_N\}$, where $Utt_i$ represents $i$-th utterance, $N$ is the number of causal variables. We adopt an iterative prompt instruction wherein the LLM's predictions are gradually corrected through feedback instructions. The causal structure predicted by the LLMs falls under model $\mathcal{V}$, while the causal representation computed through utterrance representation belongs ti model $\mathcal{U}$. Our iterative prompt instruction is as follows: firstly, the LLM is instructed to answer the complete the causal structure (model $\mathcal{V}$); secondly, the utterances' causal representation (model $\mathcal{U}$) is obtained either via pre-trained models or LLMs; finally, the differences in similarity matrices of model $\mathcal{V}$ and model $\mathcal{U}$ are fed back, and the LLM is instructed to execute the prior steps under the acknowledgement of this difference until no difference exists between the two similarity matrices. What distinguishes our LLM implementation to the SSM implementation is that model $\mathcal{U}$ is unlearnable and fixed in LLM, whereas the SSM permits both $\mathcal{V}$ and model $\mathcal{U}$ to be learnable.

Eventually, this leads to the discovery of the correct causal relationship. The steps are as follows:

**Step 1 (Prediction Causal Relationship)**: Calculating the causal relationship between any two utterances within a given dialogue, the prompt instruction employs input accompanied by an example. The specific text is as follows:

*"You are assuming the role of a researcher capable of distinguishing between causation and correlation, charged with the task of recognizing the causal relationships among individual utterances within a given dialogue. We prescribe that the judgment of causation between two utterances is based on whether the former is the intended target of the latter's response. Whereas, correlation is gauged on whether the two share similar topics or vocabulary. The following is an example:*

*Example:*

*Dialogue:*

*'1. Hazel drank too much champagne at the party.*

*2. Oh my goodness! That sounds like quite an eventful party.*

*3. Well, drinking too much alcohol can have many negative effects on the body.*

*4. Oh no, I can imagine Hazel waking up with a massive headache tomorrow.'*

*Question 1: Is there a causal relationship from utterance 1 to utterance 2?*

*Answer 1: Yes.*

*Question 2: Is there a causal relationship from utterance 1 to utterance 3?*

*Answer 2: Yes.*

*Question 3: Is there a causal relationship from utterance 1 to utterance 4?*

*Answer 3: Yes.*

*Question 4: Is there a causal relationship from utterance 2 to utterance 3?*

*Answer 4: No.*

*Question 5: Is there a causal relationship from utterance 2 to utterance 4?*

*Answer 5: No.*

*Question 6: Is there a causal relationship from utterance 3 to utterance 4?*

*Answer 6: Yes.*

*Given the above example, with its associated questions and answers, consider the following dialogue:*

*Dialogue:*

*'1. Charlotte has no idea how to avoid massive estate taxes.*

*2. Estate taxes are a topic of concern for many people in various countries.*

*3. So, does anyone else have any knowledge or ideas on how to reduce estate taxes?*

*4. Oh, that reminds me of a story about my uncle.'*

*Question 1: Is there a causal relationship from utterance 1 to utterance 2?*

*Question 2: Is there a causal relationship from utterance 1 to utterance 3?*

*Question 3: Is there a causal relationship from utterance 1 to utterance 4?*

*Question 4: Is there a causal relationship from utterance 2 to utterance 3?*

*Question 5: Is there a causal relationship from utterance 2 to utterance 4?*

*Question 6: Is there a causal relationship from utterance 3 to utterance 4?"*

**Step 2 (Calculating $Sim^s$):** Based on the results from the Step 1, the similarity matrix $Sim^s$ of the causal structures under different views is computed according to Equation 18. This similarity matrix includes only two binary values: $0$ and $1$. Note that intervention here, according to Definition 2, involves the direct elimination of intervened utterances. Taking $do_g(1, 2)$ as an example, neither the

first nor the second utterance is included in the input. Consequently, the resulting similarity matrix, $Sim \in \mathbb{R}^{(N-2)*(N-2)}$. This applies to both $Sim^r$ and $Sim^s$.

**Step 3 (Calculating $Sim^r$)**: In calculating the similarity matrix $Sim^r$ for causal representations, we explored two distinct computational methods. 1), computation is conducted through LLMs. The prompt text used is similar to that in Step 1, but substitutes the question *"Is there a causal relationship from utterance A to utterance B?"* with *"Is there a correlation relationship between utterance A and utterance B?"*. 2), using a pre-trained model such as RoBERTa, we calculate the deep representations of two utterances and then compute their similarity using cosine similarity via Equation 17.

**Step 4 (Inconsistency Feedback)**: We compare the each pair of similarity matrices $Sim^r$ and $Sim^s$ obtained from differing views. If a condition occurs where $Sim^r_{i,j} = 1$ while $Sim^s_{i,j} = 0$, it can be inferred that *"there is no common cause between the i-th utterance and the j-th utterance, and the i-th utterance should not have a causal relationship to the j-th utterance."* On the contrary, if a situation arises where $Sim^r_{i,j} = 0$ and $Sim^s_{i,j} = 1$, we can assert that *"there is a common cause between the i-th utterance and the j-th utterance, and the i-th utterance should have a causal relationship to the j-th utterance."* If 'i' refers to the first utterance, no response will be given to the clause relevant to the *'common cause.'* An example of this prompt instruction is as follows:

> *"After verification, there is no common cause between the second utterance and the third utterance, and the second utterance should not have a causal relationship with the third utterance, and there is no common cause between the third utterance and the fourth utterance, and the third utterance should not have a causal relationship with the fourth utterance. Please re-answer based on these circumstances."*

**Recursive Process**: The iterative algorithm is summarized in Algorithm 1. Steps 2 to 4 will continuously loop. The end condition is reached once all instances of $Sim^r$ and $Sim^s$ across all views are identical. The causal relationship output by the LLM during the final loop represents the final results. Step 3 represents the 'Augment' module mentioned in Section 4.1 while Step 4 embodies the 'Consistency check'. The overall objective of the instruction is to enable the LLM to identify causal relationships between utterances without causal labels.

Compared to the SSM example in Appendix C.1, the LLM-based example relies significantly on the accuracy of $Sim^r$. Therefore, it often leads to a failure in achievement of desired causal relationships. (We show the experiment results in Figure 7)

---

**Algorithm 1** Iterative Prompt Instruction

---

**Require**: A dialogue text $D = \{Utt_1, Utt_2, \ldots, Utt_N\}$, a set of matrices $Sim^s\_Set = \{Sim^s_{i,j}, i, j \in N \text{ and } i \leqslant j\} = \emptyset$, a set of matrices $Sim^r\_Set = \{Sim^r_{i,j}, i, j \in N \text{ and } i \leqslant j\} = \emptyset$, and $input\_prompt$ as shown in Step 1.

**Ensure**: Causal relation adjacency matrix $C \in \mathbb{R}^{N*N}$ in where $Sim^s\_Set$ is consistent with $Sim^r\_Set$ (Both $\neq \emptyset$).

> **procedure** INTERVENTION $(I_* = \{do_{i,j}\})$ $(i, j \in N \text{ and } i \leqslant j)$
> **while** $Sim^s\_Set = Sim^r\_Set \neq \emptyset$ **do**
>> Predict $C$ via LLM according to $input\_prompt$.
>> **for** each view $do_{i,j}$ in $I_*$ **do**
>>> Calculate $Sim^s_{i,j}$ via Step 2.
>>> $Sim^s\_Set = Sim^s\_Set \cup Sim^s_{i,j}$
>>> Calculate $Sim^r_{i,j}$ via Step 3.
>>> $Sim^r\_Set = Sim^r\_Set \cup Sim^r_{i,j}$
>> **end for**
>> Replace the $input\_prompt$ with the results about inconsistent pairs between $Sim^s\_Set$ and $Sim^r\_Set$ via Step 4.
> **end while**
> **return** $C$

---

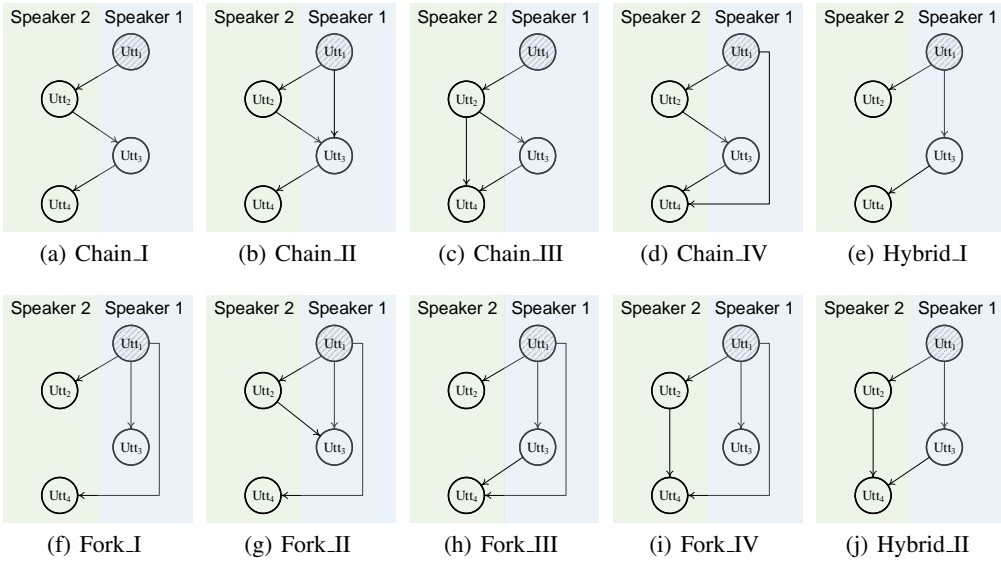

Figure 6: 10 skeletons (structures) in *Causalogue* Dataset.

# D   DETAILS ABOUT *Causalogue* DATASET

## D.1   ATTRIBUTES

**Causal Variables**: We treat each dialogue as a sample, comprised of 4 utterances, which we define as 4 causal variables. Further, the first and third utterances originate from the same speaker, defined as $speaker1$. Similarly, the second and fourth utterances are from another individual, defined as $speaker2$.

**Skeletons**: We have designed 10 types of causal skeletons (structures) in the dataset as shown in Figure 6, listed as follows:

Chain_I: This is the most basic chain structure, serving as the prototype for Chain_II-IV models. It contains three causal relationships: $Utt_1 \rightarrow Utt_2$, $Utt_2 \rightarrow Utt_3$, and $Utt_3 \rightarrow Utt_4$, representing dialogues where two speakers interact sequentially.

Chain_II: Building upon Chain_I, this model includes an additional causal relationship from $Utt_1 \rightarrow Utt_3$, indicating that $Utt_3$ considers not just the effect from $Utt_2$ but also from $Utt_1$.

Chain_III: Building upon Chain_I, this model introduces a causal link $Utt_2 \rightarrow Utt_4$, suggesting $Utt_4$ takes into account the effect of both $Utt_3$ and $Utt_2$.

Chain_IV: Building upon Chain_I, this model creates an additional causal connection $Utt_1 \rightarrow Utt_4$, indicating $Utt_4$ considers the effects from both $Utt_3$ and $Utt_1$—the two utterances by the one speaker.

Fork_I: This is the most basic fork structure, serving as the prototype for Fork_II-IV models. It includes three causal relationships: $Utt_1 \rightarrow Utt_2$, $Utt_1 \rightarrow Utt_3$, and $Utt_1 \rightarrow Utt_4$, representing situations where two speakers alternately respond to $Utt_1$ with different independent replies.

Fork_II: Building upon Fork_I, this model adds a causal relationship $Utt_2 \rightarrow Utt_3$, representing $Utt_3$'s response to not only $Utt_1$ but also $Utt_2$.

Fork_III: Building upon Fork_I, this model introduces a causal link $Utt_3 \rightarrow Utt_4$, signifying that $Utt_4$ is not merely an independent response to $Utt_1$ but a combined reply to both $Utt_1$ and $Utt_3$.

Fork_IV: Building upon Fork_I, this model incorporates a new causal relationship $Utt_2 \rightarrow Utt_4$, indicating that $Utt_4$ responds to both $Utt_2$ and $Utt_1$.

Hybrid_I: A combination of the chain and fork structures, where the chain structure runs $Utt_1 \rightarrow Utt_3 \rightarrow Utt_4$, and the fork structure is $Utt_2 \leftarrow Utt_1 \rightarrow Utt_3$.

Hybrid_II: On the basis of Hybrid_I, this introduces an additional chain structure, $Utt_1 \rightarrow Utt_2 \rightarrow Utt_4$. This model also results in a collider structure with $Utt_2 \rightarrow Utt_4 \leftarrow Utt_3$.

**Sample**: We consider a dialogue as a sample, with each sample comprising 4 utterances representing 4 causal variables. Each sample corresponds to one of the 10 causal skeletons outlined above, annotating whether a causal relationship exists between any two utterances. Due to Hypothesis 1, our labels only consider forward-causal relationships. An example of a Chain_III sample is shown as follows:

> *"causal_type": "Chain_III",*
>
> *"clause": {"1": "Your bill is 19.", "2": "Before I pay the bill, I have to express my dissatisfaction with the service I received tonight.", "3": "I'm so sorry to hear that but I don't know what happened.", "4": "Specifically, It's understandable to feel frustrated when something unexpected happens like spilling red wine on your clothes."},*
>
> *"dia_id": 1,*
>
> *"label": {"1": "0,0,0,0", "2": "1,0,0,0", "3": "0,1,0,0", "4": "0,1,1,0"}*

In the given example, the $Utt_4$ serves as a response to the $Utt_3$, while simultaneously attach to the speaker's $Utt_2$—thereby rendering both the $Utt_2$ and $Utt_3$ as causes to the $Utt_4$. Indeed, during the generation process of the $Utt_4$, we made sure to inform GPT-4 of the existence of $Utt_2$ and $Utt_3$.

## D.2 CREATION PROCESS

We utilized the API interface of GPT-4 [2] to defined the following variables: "*role*", which has three types - "*system*", "*user*", and "*assistant*". Here, "*system*" represents the background or a prior settings, while "*user*" and "*assistant*" are defined as speakers with two different identities. Additionally, the first utterance is pre-set. Hence, creating a dialogue requires a given combination: a fixed *first_utterance*, a specified *system* information, and a setting which previous utterances are considered. We have a total of 149 *first_utterance* options, and there are as many as 278,867 combinations of *first_utterance* and *system* settings (our final samples only number in the 1638, to preserve the diversity and distinctiveness of our dialogues). What follows is an example of generating the third utterance in the skeleton of ChainII:

> {"role": "system", "content": "You are Peter, you have promised to go to a Chinese Opera with your daughter, so you want to have dinner with your friends in next Sunday." }
>
> {"role": "assistant", "content": "Yes. Sunday sounds fine. What time?" (pre-set Utt_1)}
>
> {"role": "user", "content": Utt_2}

Upon creation, the samples are initially auto-annotated based on their designed labels, and then manually verified to ensure their validity. Our manual verification employed two annotators, who demonstrated proficient English understanding and communication skills, possessing sufficient knowledge about causality. The annotation consistency between these two annotators was tested through 833 samples, achieving a kappa coefficient of 0.92.

During the annotation process, if a sample was labelled differently by the two annotators, that sample was considered to possess an ambiguous causal relationship and thus was excluded from the final dataset. Only samples that were consistently labelled by both annotators were ultimately accepted.

Furthermore, to guarantee the freedom of manual annotation, we allowed the annotators to label structures that fell outside the predefined 10 causal structures. Specifically, we only requested annotators to judge whether any two utterances (satisfying Hypothesis 1) have a causal relationship, allowing them some discretion, which inevitably produced samples not belonging to the 10 causal structures. We classified these as the "Other" category.

---

[2] https://platform.openai.com/docs/models/gpt-4

The accuracy of labels was significantly improved after the manual annotation process. However, considering that the unverified samples might be utilized for other research areas, such as the ability of LLMs to focus on context, we have released two versions of the datasets, as demonstrated in Table 1. "Small" signifies samples that have been manually checked as correctly labeled, while "large"refers to all samples generated by GPT-4 without manual verification. We do not recommend considering the "large" version when undertaking causality-relarted work. Likewise, we have not taken it into our experiments.

# E    DETAILS ABOUT DATASETS, METRICS, BASELINES, AND IMPLEMENTATION

## E.1    DATASETS

The *Causalogue* dataset has already been discussed in Appendix D. Hence, this section primarily focuses on the remaining real-world datasets. Their data splits and specific $N$-folds validation setups for SSM are exhibited in Table 5. Among them, RECCON, DD, MELD, EmoryNLP and IEM are text datasets, and GTEA, 50salads, and Breakfast are video datasets. As for LLM, we only randomly select 400 samples from *Causalogue* and RECCON datasets, respectively. Their overviews and prevalent metrics are detailed below.

Table 5: Statistics on Datasets

| Dataset | Train | Valid | Test | Folds |
|---|---|---|---|---|
| Causalogue | 1338 | 100 | 200 | 10 |
| RECCON | 833 | 47 | 225 | 10 |
| DD | 11118 | 1000 | 1000 | 5 |
| MELD | 1038 | 114 | 280 | 5 |
| EmoryNLP | 713 | 99 | 85 | 5 |
| IEM | 100 | 20 | 31 | 5 |
| GTEA | 19 | 2 | 7 | 10 |
| 50salads | 36 | 4 | 10 | 10 |
| Breakfast | 1314 | 146 | 252 | 10 |

### E.1.1    EMOTION-CAUSE PAIR EXTRACTION (ECPE) TASK

**RECCON** (Poria et al., 2021): The first dataset for emotion cause recognition of conversation including RECCON-DD and RECCON-IE (emulating an out-of-distribution generalization test). RECCON-DD includes 5380 labeled ECPs and 5 cause spans (*no-context*, *inter-personal*, *self-contagion*, *hybrid*, and *latent*).

### E.1.2    EMOTION RECOGNITION IN CONVERSATION (ERC) TASK

**DD** (Li et al., 2017): A Human-written dialogs dataset with 7 emotion labels (*neutral*, *happiness*, *surprise*, *sadness*, *anger*, *disgust*, and *fear*). We follow  Chen et al. (2023b) to regard utterance turns as the speaker turns.

**MELD** (Poria et al., 2019): A multimodel ERC dataset with 7 emotion labels as the same as DD.

**EmoryNLP** (Zahiri & Choi, 2018): A TV show scripts dataset with 7 emotion labels (*neutral*, *sad*, *mad*, *scared*, *powerful*, *peaceful*, *joyful*).

**IEM** (Busso et al., 2008): A multimodel ERC dataset with 9 emotion labels (*neutral*, *happy*, *sad*, *angry*, *frustrated*, *excited*, *surprised*, *disappointed*, and *fear*). However, models in ERC field are often evaluated on samples with the first six emotions due to the too few samples of the latter three emotions. 20 dialogues for validation set is following (Chen et al., 2023b).

### E.1.3    TEMPORAL ACTION SEGMENTATION (TAS) TASK

**GTEA** (Fathi et al., 2011) Georgia Tech Egocentric Activities is comprised of 28 videos captured from a first-person perspective. It documents 7 different daily activities performed by 4 test actors, therefore, the dataset is partitioned into four 4 based on the actors. Each video contains approximately 20 fine-grained instances, with each video divided by action segments as labels.

**50salads** (Stein & McKenna, 2013) A cooking dataset includes 50 videos highlighting the complete process of salad preparation undertaken by 25 people, with each video housing between 9,000 to 18,000 RGB frames and containing 17 action class labels. Each video, named after the complete process of salad making by an individual, is segregated into 5 groups.

**Breakfast** A cooking action dataset consists of 10 cooking activities performed by 52 different actors at various kitchen locations. It encompasses 1,989 videos and offers over 77 hours of content. Each video is characterized by a sub-cooking activity accomplished by an actor; the complete preparation process comprises 20-30 such action segments. As the largest among the mentioned datasets, it is divided into 4 groups.

### E.2 EVALUATION METRICS

#### E.2.1 CAUSAL DISCOVERY (CD) TASK

The CD task was evaluated on the *Causalogue* dataset, a brand new dataset released by us. In our experiments, we endeavored to assess three outcomes: the accuracy of causal graphs, the accuracy of causal representations, and the consistency between causal graphs and representations. Consequently, we employed AUROC and Hamming Distance (HD) to measure causal graphs, AUROC and F1 scores for causal representation evaluation, and MSE and $1-$AUROC for measuring the distance of inconsistencies. These metrics are common and well-accepted. Simultaneously, for each outcome, we ensured two different metrics to comprehensively evaluate the performance.

#### E.2.2 EMOTION-CAUSE PAIR EXTRACTION (ECPE) TASK

We continue to empoly the F1 score as the evaluation metric, as initially proposed in Poria et al. (2021). This metric is broadly accepted and utilized in current research works (Li et al., 2023a;b).

#### E.2.3 EMOTION RECOGNITION IN CONVERSATION (ERC) TASK

Similarly to ECPE task, We continue to empoly the F1 score as the evaluation metric, as initially proposed in Shen et al. (2021). This metric is broadly accepted and utilized in current research works (Chen et al., 2023b; Zhang et al., 2023a; Shi & Huang, 2023).

#### E.2.4 TEMPORAL ACTION SEGMENTATION (TAS) TASK

Commonly used metrics include frame-level accuracy (Acc), segmental edit distance (Edit), and segmental F1 scores with different overlapping threshold k (F1@k) ($k = \{10, 25, 50\}$). Moreover, to evaluate the causal consistency of the segmentation results, we proposed an additional causal edit distance (C-Dis) to measure the dissimilarity between the adjacency matrix and the ground truth. For the final segmentation results, we constructed causal adjacency matrices $\hat{C} \in \mathbb{R}^{T*T}$ and ground truth matrices $C \in R^{T*T}$, based on the constraints in consistent mapping condition and calculated the dissimilarity between them.

$$C - Dis := num(\hat{C}_{i,j} \neq C_{i,j}) for\ i, j = 1, 2, \dots, T \tag{20}$$

A lower causal edit distance indicates that the causal relationship at the frame-level has less dissimilarity with the ground truth, demonstrating stronger learning ability with causal representation in the model, and hence a higher level of causal consistency in the segmentation results.

### E.3 BASELINES

#### E.3.1 BASELINES ON CD TASK

SSM

**ACD**: leverages shared dynamics to learn to infer causal relationships from multi-skeleton time-series data via a single, amortized model.

**DAG-GNN**: leverages SCM to construct a gnn-based variational model adopting independent noise $E$ as latent viariable.

**ACCD**: discover causal relationships in multi-value data via designing a common skeleton and generating a substitute for independent noise.

**biCD**:proposes a dynamic variational inference model leveraging the causal strength instead of independent noise as the latent variable to construct ELBO for indefinite data.

**DisC**: designs a new method for intervention in deep models, combining causal patterns with different shortcuts to achieve the goal of intervention in causal nodes.

**DIR**:distinguishes between positive and negative samples after intervention by designing a dynamic loss function, Similar to the DisC thereby effectively intervening in the causal pattern.

Since DisC and DIR do not have complete causal discovery models, we incorporate their intervention modules into DAG-GNN (namely, DAG-DisC and DAG-DIR) to demonstrate their intervention strategies for indefinite data.

LLM

**Zero-shot and Zero-shot-CoT**: proposes a new prompt paradigm like "Let's think step by step" which is task-agnostic and does not need input-output demonstrations.

**Auto-CoT**: proposes an auto prompt method which could cluster the samples first and then select an example for prompt text.

### E.3.2 BASELINES ON ECPE TASK

**EDKA-GM**:introduces an experiencer identification task and present a document-level heterogeneous graph network for capturing global experiencer information to enrich experiencer-based cross-clause association.

**seF**:includes two main components: core clause selector and emotion-cause pairs extractor to jointly extract emotion-cause pairs.

### E.3.3 BASELINES ON ERC TASK

**DAG-ERC**: proposes a gnn&rnn-based model to learn the relationship of different speakers and sequential information.

**DualGAT**:introduces Dual Graph Attention networks to concurrently consider the complementary aspects of discourse structure and speaker-aware context.

**MultiEMO**:proposes a novel attention-based correlation-aware multimodal fusion framework effectively integrating multimodal cues by capturing cross-modal mapping relationships across textual, audio and visual modalities.

### E.3.4 BASELINES ON TAS TASK

**MS-TCN**: This is the first method to introduce a multi-stage action segmentation framework based on Temporal Convolutional Networks (TCN). Each stage inputs the initial prediction output from the preceding one for further modification and adjustment.

**MS-TCN++**: On the foundation of MS-TCN, this method introduces a dual dilated layer, implementing parameter sharing and optimizing segmentation performance.

**ASRF**: This method proposes an improved technique based on MS-TCN, composed of a long-term feature extractor and two branches: the Action Segmentation Branch (ASB) and the Boundary Regression Branch (BRB).

**CETNet**: Leveraging Transformer, this method connects every layer of convolutional feature mapping in the encoder with a group of features generated through self-attention in the decoder.

**C2F**: Utilizing TCN, this method puts forward a novel temporal encoder-decoder to tackle the sequence fragment issue. Its decoder conforms to a coarse-to-fine structure with multi-timescale implicit integration.

### E.4 Implementation Details

### E.4.1 The Model on CD Task

In our Experiments, we utilized RoBERTa-base (768) as our pre-trained model for generating word embeddings in the SSMs. Throughout the training process, a learning rate of 1e-5 was set, with the batch size and epochs set to 16 and 50, respectively. The dimension of the hidden layers within the network was also set to 768. The entire training procedure was conducted on a NVIDIA GEFORCE RTX 3090 graphics processing unit.

### E.4.2 The Model on ECPE Task and ERC Task

In the word embedding, we adopt the affect-based pre-trained features[3] proposed by Shen et al. (2021) for all baselines and models.

In the hyper-parameters, we follow the setting of Chen et al. (2023b) in the ERC task. Moreover, in the ECPE, the learning rate is set to 3e-5, batch size is set to 32, and epoch is set to 60. Further in our approach, hidden size of GNN is set to 300, and dropout rate is 0.3. All experiments were conducted on a NVIDIA GEFORCE RTX 3090 for both training and testing.

### E.4.3 The Model on TAS Task

We employed the features extracted from I3D (Carreira & Zisserman, 2017) as the input for our model. To avoid random bias, we applied our augmented approach across different backbones while retaining the seed setup from their original studies, ensuring that the specific training epochs are consistent with the backbones. All experiments were conducted on a NVIDIA GEFORCE RTX 3090 for both training and testing. We set the learning rate to 0.0005, established a weight decay of 0.001, and utilized Adam as the optimizer. To enhance the training efficiency and avert degenerate matrices during whitening, we set the batch size of the frame images for a video segment to 512. We followed the recommendation in Ermolov et al. (2021) to further subdivide the batches during the whitening process, setting the sub-batch size to 128.

## F Results on LLMs

In our LLMs experiments, we devised a simple task: predicting the existence of a causal relationship between any two utterances (yes or no) to gauge the LLM's capability for causal reasoning. Table 6 illuminates some interesting conclusions - existing prompting methods is difficult to yield effective outcomes for this task. For instance, the "step by step" thinking guided by the CoT approach tends to make LLMs involve many correlation-based responses. The cluster approach of Auto-CoT also contributes meaninglessly when the samples are too similar. Conversely, our iterative prompts instruction enable LLMs to uncover causal inconsistencies in its previous answers, thereby allowing for self-correction. This self-supervised idea appears to impose the LLMs with a capability of "reasoning".

Table 6: The F1 score of causal relationship recognition of prompt Models. $\text{Ours}^1_{LLM}$ represents that calculating $Sim^r$ via LLM while $\text{Ours}^1_{LLM}$ represents it via RoBERTa pre-trained model.

| Model | Causalogue | RECCON |
|---|---|---|
| Zero-Shot | 0.61 | 0.52 |
| Zero-Shot-CoT | 0.58 | 0.51 |
| Auto-CoT | 0.62 | 0.51 |
| $\text{Ours}^1_{LLM}$ | 0.74 | 0.66 |
| $\text{Ours}^2_{LLM}$ | 0.72 | 0.69 |

However, the accuracy of our proposed iterative propmt is substantially dependent on the precision of $Sim^r$. An incorrect $Sim^r$ can lead the correct results to be modified. Furthermore, different from the SSMs that could make $Sim^r$ and $Sim^s$ be trained together, the LLM does not provide a learnable module to refine $Sim^r$, only leading to the $Sim^s$ close to the fixed $Sim^r$. Figure 7 (a, b, c) represents the $Sim^r$ calculations obtained through three different measures, where 'c' derives from labels, providing 100% accurate. The red dashed line in the lineplot denotes the F1 of $Sim^r$. Our method

---

[3] https://drive.google.com/file/d/1R5K_2PlZ3p3RFQ1Ycgmo3TgxvYBzptQG/view?usp=sharing

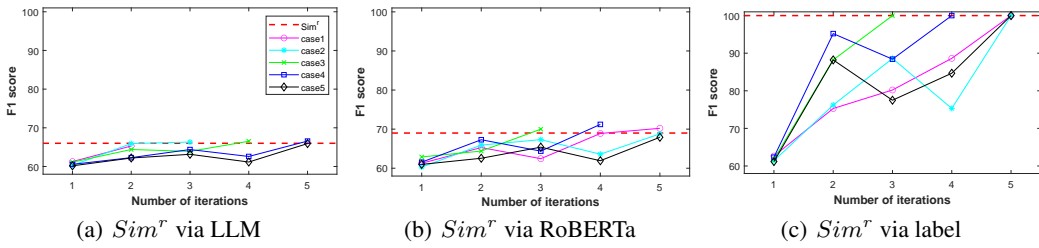

(a) $Sim^r$ via LLM  (b) $Sim^r$ via RoBERTa  (c) $Sim^r$ via label

Figure 7: The upper bound of $Sim^r$ and F1 scores of 5 cases in different $Sim^r$ calculating measures

was applied across 5 cases, using each of 3 measures respectively. The results consistently converge around the $Sim^r$ which indicates that $Sim^r$ is the upper bound of identifying causal relationships.

Finally, in order to demonstrate the specific question and answer process of our iterative prompt, we provide a complete case as follows.

---

*\*\*user\*\*: You are assuming the role of a researcher capable of distinguishing between causality and correlation, charged with the task of assessing the causal relationships among individual utterances within a given dialogue. We prescribe that the judgment of causality between two sentences is based on whether the former is the intended target of the latter's response. Whereas, correlation is gauged on whether the two share similar topics or vocabulary. The following is an example:*

*Example:*

*Dialogue:*

*Hazel drank too much champagne at the party. Oh my goodness! That sounds like quite an eventful party. Well, drinking too much alcohol can have many negative effects on the body. Oh no, I can imagine Hazel waking up with a massive headache tomorrow.*

*Question 1: Is there a causal relationship from utterance 1 to utterance 2? Answer 1: Yes.*

*Question 2: Is there a causal relationship from utterance 1 to utterance 3? Answer 2: Yes.*

*Question 3: Is there a causal relationship from utterance 1 to utterance 4? Answer 3: Yes.*

*Question 4: Is there a causal relationship from utterance 2 to utterance 3? Answer 4: No.*

*Question 5: Is there a causal relationship from utterance 2 to utterance 4? Answer 5: No.*

*Question 6: Is there a causal relationship from utterance 3 to utterance 4? Answer 6: Yes.*

*Given the above example, with its associated questions and answers, consider the following dialogue:*

*Dialogue:*

*Charlotte has no idea how to avoid massive estate taxes. Estate taxes are a topic of concern for many people in various countries. So, does anyone else have any knowledge or ideas on how to reduce estate taxes? Oh, that reminds me of a story about my uncle.*

*Question 1: Is there a causal relationship from utterance 1 to utterance 2?*

*Question 2: Is there a causal relationship from utterance 1 to utterance 3?*

*Question 3: Is there a causal relationship from utterance 1 to utterance 4?*

*Question 4: Is there a causal relationship from utterance 2 to utterance 3?*

*Question 5: Is there a causal relationship from utterance 2 to utterance 4?*

*Question 6: Is there a causal relationship from utterance 3 to utterance 4?*

---

*\*\*chatgpt\*\*: Answer 1: Yes. Utterance 2 is a response to the topic brought up in utterance 1, specifically the issue of estate taxes.*

*Answer 2: Yes. Utterance 3 directly follows up on the issue presented in utterance 1, asking if others have knowledge about the same topic.*

*Answer 3: No. While utterance 4 may be about a related issue, there is no explicit causal relationship based on the given information.*

*Answer 4: Yes. Utterance 3 is a continuation of the discussion about estate taxes, which was the topic of utterance 2.*

*Answer 5: No. Utterance 4 does not indicate a response or direct continuation of the topic presented in utterance 2.*

*Answer 6: Yes. The story mentioned in utterance 4 could be a response to the question asked in utterance 3.*

*\*\*user\*\*: After verification, the first utterance should have a causal relationship with the fourth utterance, and there is no common cause between the third utterance and the fourth utterance, and the third utterance should not have a causal relationship with the fourth utterance, and there is no common cause between the second utterance and the third utterance, and the second utterance should not have a causal relationship with the third utterance. Please re-answer based on these circumstances.*

*\*\*chatgpt\*\*: With these new circumstances:*

*Answer 1: Yes. Utterance 2 is a response to the topic brought up in utterance 1, specifically the issue of estate taxes.*

*Answer 2: No. Even though utterance 3 follows utterance 1, we are told that there's no causal relationship between them.*

*Answer 3: Yes. Based on your new information, utterance 4 now has a causal relationship with utterance 1.*

*Answer 4: No. Taking into account the revised situation, there's no causal link between utterance 2 and utterance 3.*

*Answer 5: No. Utterance 4 does not indicate a response or direct continuation of the topic presented in utterance 2.*

*Answer 6: No. According to the new context, there's no causal relationship between utterance 3 and utterance 4.*

*\*\*user\*\*: After verification, the first utterance should have a causal relationship with the fourth utterance. Please re-answer based on these circumstances.*

1014

*\*\*chatgpt\*\*: Given the new specification:*

*Answer 1: Yes. Utterance 2 is a response to the topic brought up in utterance 1, specifically the issue of estate taxes.*

*Answer 2: Yes. Utterance 3 directly follows up on the issue presented in utterance 1, asking if others have knowledge about the same topic.*

*Answer 3: Yes. Based on your new information, utterance 4 now has a causal relationship with utterance 1 (Charlotte's lack of knowledge about estate taxes somehow triggers the story about the speaker's uncle).*

*Answer 4: Yes. Utterance 3 is a continuation of the discussion about estate taxes, which was the topic of utterance 2.*

*Answer 5: No. Utterance 4 does not indicate a response or direct continuation of the topic presented in utterance 2.*

*Answer 6: Yes. Under normal circumstances, the story mentioned in utterance 4 can be perceived as a response to the question asked in utterance 3. However, if we strictly adhere to the new context provided("no common cause between the third utterance and the fourth utterance"), then the answer would be: No. There is no causal relationship between utterance 3 and utterance 4.*

---

***user**: After verification, there is no common cause between the second utterance and the third utterance, and the second utterance should not have a causal relationship with the third utterance, and there is no common cause between the third utterance and the fourth utterance, and the third utterance should not have a causal relationship with the fourth utterance. Please re-answer based on these circumstances.*

---

***chatgpt**: Given the additional constraints:*

*Answer 1: Yes. Utterance 2 is a response to the topic brought up in utterance 1, specifically the issue of estate taxes.*

*Answer 2: Yes. Despite the absence of causal links to utterances 2 and 4, utterance 3 directly follows up on the issue presented in utterance 1, asking if others have knowledge about the same topic.*

*Answer 3: Yes. Based on your new information, utterance 4 now has a causal relationship with utterance 1 (Charlotte's lack of knowledge about estate taxes reminded the speaker of a story about their uncle).*

*Answer 4: No. Based on your new information, there is no causal relationship between utterance 2 and utterance 3.*

*Answer 5: No. Utterance 4 does not indicate a response or direct continuation of the topic presented in utterance 2.*

*Answer 6: No. Based on your updated context, there is no causal relationship between utterance 3 and utterance 4.*

## G    RESULTS ON ECPE, ERC, AND TAS TASKS

We assessed the performance of Ours$_{SSM}$ on three downstream tasks: ECPE, ERC, and TAS. These tasks not only represent typical instances of indefinite data (text and video), but they also embody causally related tasks. For instance, in ECPE and ERC tasks, mastering the causal relationships between utterances is vital, while in the TAS task, recognizing and effectuating transformation from intra-frame relations to intra-segment relations is crucial.

From the outcomes presented in Tables 7 and 8, Ours$_{SSM}$ exhibits a remarkable improvement when dealing with these high-level causal models. The underlying reason for this enhancement is that, under the conditions of ensured causal consistency, an increase in the accuracy of the causal model promotes enhancements in both the causal structure (C-Dis in Table 8) and causal representation (Edit in Table 8), surpassing other methods, hence improving the final results.

Table 7: Results of SSMs on ECPE and ERC tasks. 95% confidence interval shown. All evaluation metrics used in the Table were F1 scores (Appendix E.2). The backbone of Ours$_{SSM}$ are biCD and DualGAT, respectively.

| ECPE | | ERC | | | | |
|---|---|---|---|---|---|---|
| **Model** | **RECCON** | **Model** | **MELD** | **EmoryNLP** | **DD** | **IEM** |
| ACCD | $73.17_{\pm1.1}$ | ACCD | $63.81_{\pm0.11}$ | $39.54_{\pm0.12}$ | $59.53_{\pm0.01}$ | $69.17_{\pm0.15}$ |
| biCD | $74.14_{\pm0.74}$ | biCD | $63.22_{\pm0.17}$ | $38.21_{\pm0.11}$ | $59.64_{\pm0.07}$ | $67.15_{\pm0.09}$ |
| EDKA-GM | $72.14_{\pm0.93}$ | DAG-ERC | $63.65_{\pm0.05}$ | $39.02_{\pm0.13}$ | $59.33_{\pm0.01}$ | $68.03_{\pm0.15}$ |
| seF | $74.55_{\pm0.98}$ | DualGAT | $66.72_{\pm0.12}$ | $40.88_{\pm0.15}$ | $61.80_{\pm0.02}$ | $67.74_{\pm0.21}$ |
| - | - | MultiEMO | $61.23_{\pm1.26}$ | $37.14_{\pm0.11}$ | $57.46_{\pm0.01}$ | $64.41_{\pm0.16}$ |
| Ours$_{SSM}$ | $\mathbf{76.89}_{\pm1.21}$ | Ours$_{SSM}$ | $\mathbf{67.79}_{\pm0.18}$ | $\mathbf{40.95}_{\pm0.08}$ | $\mathbf{62.57}_{\pm0.01}$ | $\mathbf{69.81}_{\pm0.26}$ |

Table 8: Results of SSMs on TAS task. All evaluation metrics used in the Table were introduced in Appendix E.2.4. The backbone of Ours$_{SSM}$ is CETnet.

| Model | GTEA | | | | | | 50salads | | | | | | Breakfast | | | | | |
|---|---|---|---|---|---|---|---|---|---|---|---|---|---|---|---|---|---|---|
| | F1@{10, 25, 50} | | | Edit | Acc | C-Dis | F1@{10, 25, 50} | | | Edit | Acc | C-Dis | F1@{10, 25, 50} | | | Edit | Acc | C-Dis |
| MSTCN++ | 82.3 | 83.6 | 71.9 | 79.8 | 77.6 | 8.4 | 79.4 | 77.3 | 69.3 | 71.6 | 82.8 | 3.3 | - | - | - | - | - | - |
| ASRF | 85.5 | 83.8 | 73.6 | 76.9 | 74.7 | 9.0 | 80.3 | 77.4 | 67.4 | 74.2 | 77.6 | 4.9 | 69.1 | 63.4 | 50.8 | 66.6 | 63.0 | 55.8 |
| CETnet | 90.5 | 89.6 | 78.9 | 85.7 | 79.4 | 7.1 | 87.6 | 87.3 | 80.9 | 82.8 | 87.3 | 2.6 | 72.5 | 68.7 | 57 | 72.8 | 74.2 | 38.1 |
| C2F | 88 | 86.6 | 78.3 | 81.6 | **80.6** | 7.4 | 83.5 | 81.5 | 71.8 | 75.7 | 86.9 | 2.8 | 71.6 | 68.0 | 57.1 | 68.1 | 74.6 | 49.8 |
| Ours$_{SSM}$ | **91.4** | **90.2** | **80.5** | **87.2** | 79.7 | **6.9** | **88.9** | **87.6** | **81.4** | **83.1** | **88.9** | **2.5** | **78.7** | **74.9** | **63.4** | **78.3** | **75.6** | **35.4** |

To better illustrate the role of the causal model in these downstream tasks, we demonstrate two visualizations in Figures 8 and 9. Figure 8 displays a visualization of the adjacency matrix for the ECPE task, which can be equated with a causal graph, showing how the model assigns weights to the context when learning utterance relationships. Figure 8 demonstrates that the superiority of the causal method over non-causal ones lies in turning the adjacency matrix into a DAG, thus avoiding the factual error of treating earlier utterances as outcomes of latter ones. However, due to unknown causal labels, there is not a sufficiently strong constraint for causal graph, which often leaves the model uncertain about which edges in the DAG should exist. Our model mitigates this issue by using causal consistency constraints, enabling the model to identify the correct edges through contrastive learning of the causal representation and structure.

Figure 9 shows the TAS task's visualization results, illustrating that causal consistency between frames and segments can significantly reduce the exsitings of trivial segments. This harmonizes with our intuition on indefinite data: all frames within a segment share a similar causal relationship. Such a causal model goes beyond the scope of this paper, and we thoroughly discuss these extended contributions in the discussion section.

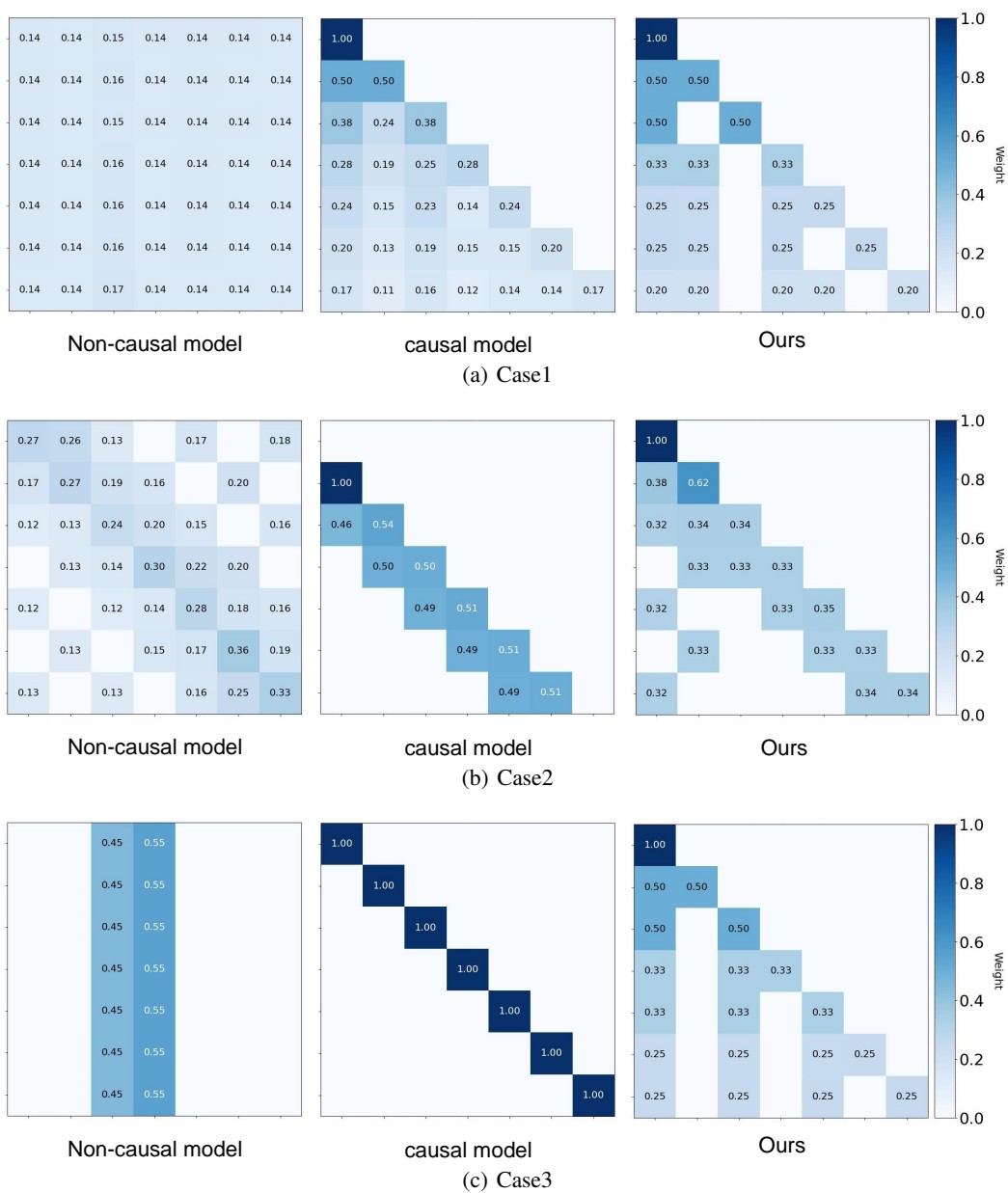

Figure 8: Visualization of adjacent matrices of 3 cases on ECPE task. Non-causal model is EDKA-GM, and we choose ACCD as causal model. The adjacent matrix is N*N representing the relationship between any two utterances.

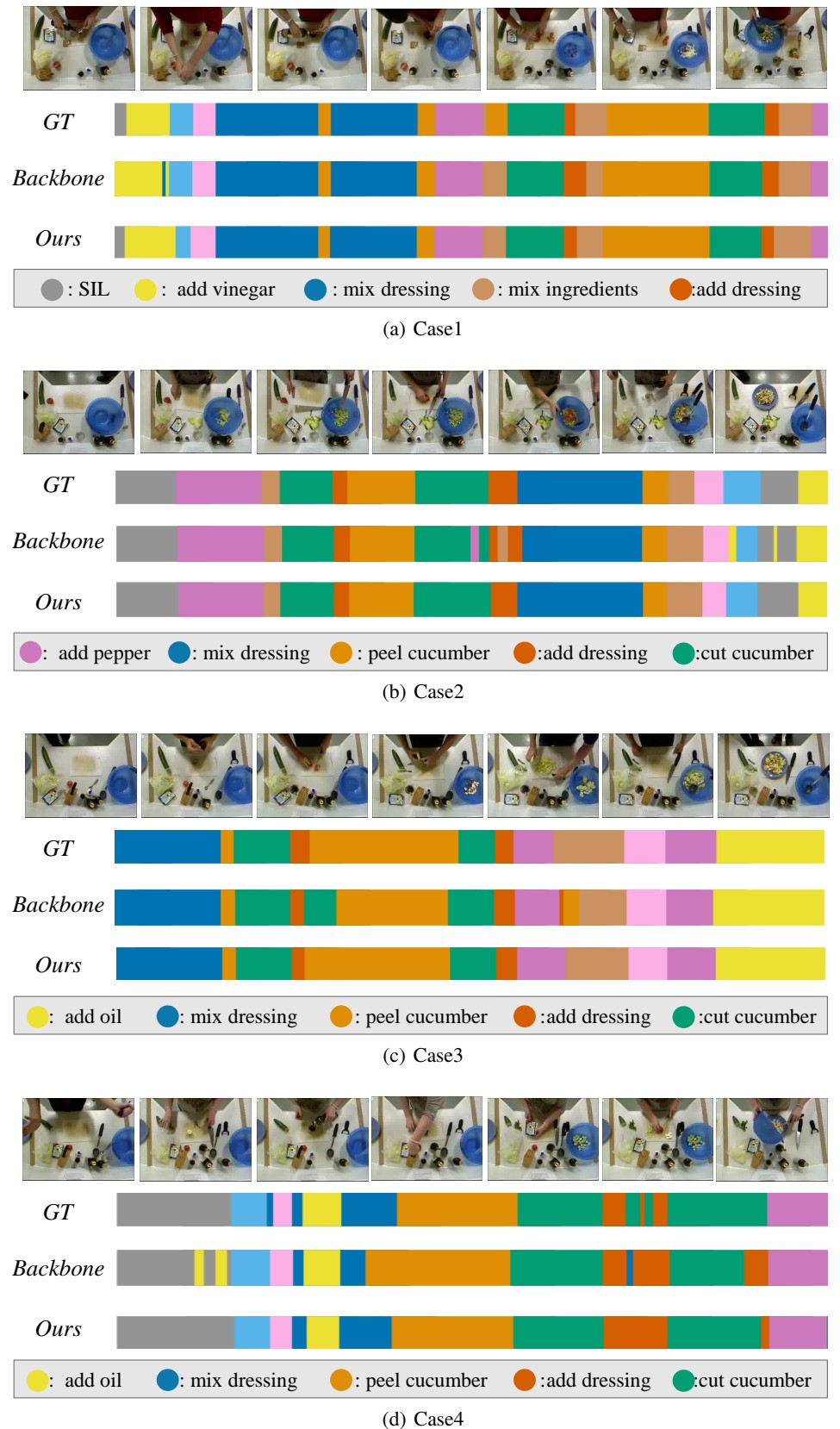

Figure 9: Visualization of results of 4 cases on 50salads dataset. GT represents the Ground Truth, Backbone we choose is the MSTCN++.

