# ADDED AND MODIFIED CONTENT FOR PAPER #575

## 1 ADDED NEW DEFINITION

**Definition 100** (Causal representation). *The causal representation $\hat{X}$ represents the computed values of causal variables when constructing a causal model, i.e., the quantified values from causal variables. Causal representations should meet the following two conditions:*

- *Correlation Condition: For any two causal variables that are not independent, their corresponding causal representations must be correlated.*

- *Causation Condition: For any two causal variables that have direct causal relationship, their causal representations should contain not only the information about their correlation but also information about the causal relationship.*

*For example, in SCM, information about the noise terms can be included, where correlation relationship can be determined by fitting or comparing measures (such as cosine similarity), and causal relationship can be determined by examining the residuals $\Sigma$ of the fitted model (Chen et al., 2023a).*

- $\Sigma_X \perp\!\!\!\perp Y, \Sigma_Y \not\!\perp\!\!\!\perp X \Rightarrow Y \rightarrow X$

- $\Sigma_X \not\!\perp\!\!\!\perp Y, \Sigma_Y \perp\!\!\!\perp X \Rightarrow X \rightarrow Y$

- $\Sigma_X \not\!\perp\!\!\!\perp Y, \Sigma_Y \not\!\perp\!\!\!\perp X \Rightarrow L \rightarrow X, L \rightarrow Y$

- $\Sigma_X \perp\!\!\!\perp Y, \Sigma_Y \perp\!\!\!\perp X \Rightarrow X \rightarrow L, Y \rightarrow L$

The difference between causal representation and ordinary deep representation lies in the "Causation Condition" mentioned in Definition 100. In general, deep representation can only meet the Correlation Condition, meaning it can only identify correlations. However, causal representation can identify not only correlations, but also causal relationships.

We noted that some reviewers also had questions about the causal structure. By causal structure, we mean causal diagrams, which we believe is a widely recognized term.

## 2 THE DETAILS ABOUT CAUSAL CONSISTENCY

Let's illustrate the concept of causal inconsistency using a simple example: Consider a simple causal structure $A \leftarrow C \rightarrow B$, from the structure, we can see that $A$ and $B$ are correlated. This is due to $P(A, B|C) = P(A|C) * P(B|C) \Rightarrow A \not\!\perp\!\!\!\perp B (A \perp\!\!\!\perp B|C)$. In the case of single-valued variables, we conduct independence tests on all samples of $A$ and $B$ to ascertain whether they are correlated. If they are, we then conclude that the causal structure and representation are consistent; otherwise, they are inconsistent. For multi-valued variables (such as deep representation satisfying Definition 100), one method to approximate this "correlation" is using cosine similarity or mean squared error (MSE). If the representations of $A$ and $B$ are similar, we also consider the structure and the representation to be consistent.

Hence, we use a similarity matrix to measure this "inconsistency." To continue with the example above, we hypothesize two similarity matrices for the structure and representation respectively, $Sim^s \in \mathbb{R}^{3*3}$ and $Sim^r \in \mathbb{R}^{3*3}$. In these, $Sim^s_{i,j} = P(i|j)$, and $Sim^r_{i,j} = cossim(i, j)$. The MSE between these two similarity matrices $Sim^s$ and $Sim^r$ is used to measure inconsistency - if the MSE is close to 0, it indicates that $Sim^s$ is approximately equal to $Sim^r$, i.e., the causal structure and the causal representation are essentially consistent, otherwise they are inconsistent.

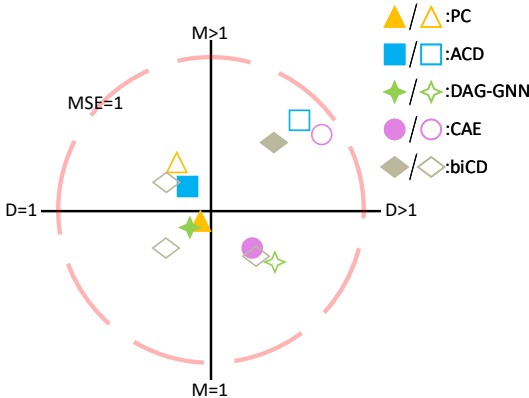

Figure 100: We compared the consistency of different methods in 3 data paradigms (if available). The consistency was represented by the MSE of the similarity matrices for structure and representation. The filled markers represent methods being in their default data forms, while the hollow markers signify that they are in extendable but non-default data forms.

## 3 MODIFIED SECTION: WHY CAUSAL INCONSISTENCY ARISES?

Figure 100 visualizes evaluation results of causal consistency via tested 5 methods: PC (Kalisch & Bühlman, 2007), ACD (Löwe et al., 2022), DAG-GNN (Yu et al., 2019), CAE (Chen et al., 2023a), and biCD (Chen et al., 2023b). They represent prevalent methods in specific data forms, respectively. Two conclusions can be obtained from Figure 100:

- The strongest causal inconsistency is found in indefinite data forms ($M > 1$, $D > 1$), while definite data ($M = 1$, $D = 1$) performs the weakest causal inconsistency.

- When existing methods are applied to non-default data forms (hollow markers), their consistency performance is always inferior to the native methods for that data form.

In addition to experimental results, we also provide comprehensive theoretical analysis for three different data paradigms:

- **Definite Data (M=1 and D=1)**: The causal strength $f$ is fixed and can be recovered through statistical properties in the data (for example, independent tests, independent component analysis, rank of covariance, etc.). Therefore, the estimated causal representation $\hat{X} = X$ (D=1), and the causal strength $\hat{f} = f$ (M=1). The subtle inconsistencies in Figure 100 arise from biases or confounding in the sampling process.

- **Semi-Definite Data (D>1 and M=1)**: According to Definition 100, there are differences between the causal representation $\hat{X}$ and the input representation $X$, therefore we can't directly optimize causal representation through $loss(\hat{X}, X)$. Fortunately, in this situation, $f$ is fixed, so we can map $\hat{X}$ to a unique $\hat{f}$ without parameters, and then optimize the process of causal discovery through $loss(\hat{f}, f)$. Thus, $\hat{f}$ is the projection of $\hat{X}$ and inherently consistent. The effectiveness of $\hat{X}$ comes from: $\hat{X} \Leftrightarrow \hat{f} = f \Leftrightarrow X$. The slight inconsistencies in Figure 100 result from biases of projection.

- **Semi-Definite Data (M>1 and D=1)**: The estimated causal strength $\hat{f}$ can be viewed as distribution determined by $X$ and encoder parameter $\varphi$, denoted as $\hat{f} = h(X, \varphi)$, and is optimized through $loss(\hat{f}, f)$. $\hat{X}$ can be estimated via inverse function $h^{-1}$, because when the causal variable does not need to be quantified into a deep representation, there exists an error $loss(\hat{X}, X)$ such that $\hat{X} = X$. From this, we can get the equivalent equation: $\hat{f} = f \Leftrightarrow X = \hat{X}$ ( $\Leftrightarrow$ is because of the ground-truth information). Thus, $\hat{f}$ and $\hat{X}$ are consistent. The minor inconsistencies in Figure 100 arise from biases existing after the convergence of the two losses.

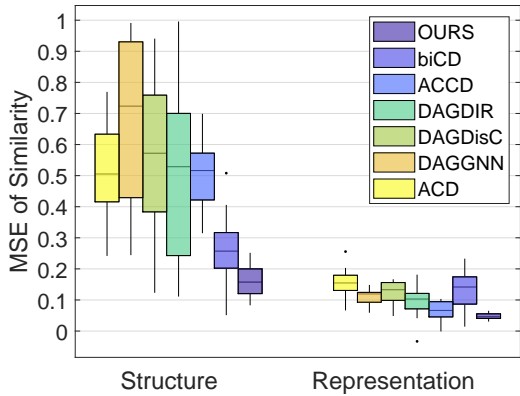

Figure 101: The boxplot showing the MSE of predicted similarity matrices to ground truth. The left clustering is the results from strucure to the ground truth, and right clustering is results from representation. The similarity matrices are computed via Equation 17 and 18, respectively.

Table 100: Results of Scalability on *Causalogue* Dataset. "$auroc_S$", "$auroc_R$", and "$auroc_C$" represent the AUROC of Causal Strucutre, Causal Representation and Causal Consistency, respectively.

| Methods | 20% Trainset | | | 40% Trainset | | | 60% Trainset | | | 80% Trainset | | | 100% Trainset | | |
|---|---|---|---|---|---|---|---|---|---|---|---|---|---|---|---|
| | $auroc_S$ | $auroc_R$ | $auroc_C$ | $auroc_S$ | $auroc_R$ | $auroc_C$ | $auroc_S$ | $auroc_R$ | $auroc_C$ | $auroc_S$ | $auroc_R$ | $auroc_C$ | $auroc_S$ | $auroc_R$ | $auroc_C$ |
| ACD | 0.15 | 0.22 | 0.12 | 0.49 | 0.45 | 0.19 | 0.67 | 0.68 | 0.31 | 0.79 | 0.85 | 0.44 | 0.84 | 0.85 | 0.51 |
| DAG-GNN | 0.08 | 0.26 | 0.16 | 0.36 | 0.44 | 0.25 | 0.49 | 0.65 | 0.33 | 0.54 | 0.88 | 0.45 | 0.56 | 0.90 | 0.50 |
| DAG-DisC | 0.07 | 0.24 | 0.08 | 0.38 | 0.39 | 0.24 | 0.57 | 0.63 | 0.31 | 0.64 | 0.81 | 0.46 | 0.68 | 0.88 | 0.52 |
| DAG-DIR | 0.10 | 0.26 | 0.14 | 0.47 | 0.48 | 0.29 | 0.51 | 0.63 | 0.28 | 0.63 | 0.84 | 0.46 | 0.67 | 0.89 | 0.51 |
| ACCD | 0.13 | 0.19 | 0.12 | 0.45 | 0.46 | 0.27 | 0.61 | 0.65 | 0.39 | 0.74 | 0.87 | 0.46 | 0.79 | 0.93 | 0.60 |
| biCD | 0.16 | 0.25 | 0.18 | **0.53** | 0.49 | 0.26 | 0.74 | 0.69 | 0.37 | 0.84 | 0.82 | 0.57 | 0.91 | 0.86 | 0.64 |
| Ours$_{SSM}$ | **0.21** | **0.44** | **0.18** | 0.52 | **0.61** | **0.35** | **0.75** | **0.79** | **0.69** | **0.88** | **0.90** | **0.89** | **0.94** | **0.94** | **0.95** |

- **Indefinite Data (M>1 and D>1)**: For multi-structure scenarios, $\hat{f} = h_1(X, \varphi)$, and for multi-value variables, $\hat{X} = h_2(X, \hat{f})$. And $D > 1$ makes $loss(\hat{X}, X)$ ineffective. Therefore, when only $loss(\hat{f}, f)$ exists, we can get $\hat{f} = f$ and $f \Leftrightarrow X$. However, we cannot guarantee $\hat{X} \Leftrightarrow \hat{f}$ or $X = \hat{X}$, thus severe inconsistencies exist.

## 4 VARIANCE OF LEARNING RESULTS

In Table 3, we only show the evaluation results between structure (graph) and representation. To further demonstrate the benefits of our SSL framework to the model, we additionally focus on the error of the similarity matrices of structure to the ground truth, and representation to the ground truth, respectively. Figure 101 shows the box plots of the errors in structure and representation. Ours$_{SSM}$ evidently outperforms other methods in terms of structure, even those specifically designed to handle multi-value data (ACCD, biCD). In addition, the variance of intervention-based methods (DAG-DisC, DAG-DIR) is extremely large, which aligns with our previous conclusion that intervening by negative examples leads to the additional variance from the batch size. On the representation side, almost all methods performed well. As indicated in our definition 100, nearly all methods can satisfy Correlation Condition, hence the error can be reduced to a certain extent. Nevertheless, the remaining stubborn error is due to "pseudo-correlation" caused by the inability to fully satisfy Causation Condition due to causal inconsistency. The significantly smaller variance of our method demonstrates that CCC can further help representation more completely determine Causation Condition.

## 5 SCALABILITY

We evaluate scalability by scaling the training set. Table 100 shows that our method performs best under any scale of datasets, especially in terms of structure. This is because, when the sample size is insufficient, intervention methods can extract more causal information contained in the samples. At the same time, the various real datasets in Appendix G also indirectly reflect our method's adaptability to datasets of different scales.