# OpenReview forum: "SSL Framework for Causal Inconsistency between Structures and Representations"
_ICLR.cc/2024/Conference — Submitted to ICLR 2024_

### Official Review · Reviewer_2DS3 · 2023-10-24

**Soundness:** 3 good
**Presentation:** 2 fair
**Contribution:** 2 fair
**Rating:** 6
**Confidence:** 3

**Summary:**

In this paper, the author proposes a SSL framework to make the causal structure be the same as causal representation by regulating their strength set. The author further introduces a dialogue dataset which could serve as a potential benchmark.

**Strengths:**

1. The author showcases implementations under different frameworks.

2. The dataset could be useful.

3. The experimental results are promising.

**Weaknesses:**

1. The presentation and the readability of the paper can be improved.

The author points out the inconsistency between causal structure and causal representation.  However, there is no formal definition of causal consistency in the paper.  In theory 1, the author gives the causal consistency condition as if it is equivalent under any intervention, these two causal models are consistent. But still, it is not a definition and there is a lack of intuition and motivation of why inconsistency is a problem to be solved.


2. The hypothesis 2 is confusing.

In images or text, people project the original data into latent space, where each dimension does not have to be entangled. Even in the feature space, they do not have to be entangled. The pixel one may not relate to pixel two.

**Questions:**

1. What is the causal inconsistency between causal representation and causal structure and why do we want to optimize it.


2.  How do you adapt your method to other datasets where the intervention cannot be done, for example, celebA.

---

> ### Author Response · Authors · 2023-11-11
> **Response to reviewer 2DS3**
>
> Thank you for your careful review. We will address your concerns in turn with the following responses:
>
> 1. About Causal Consistency:
>
> We recognize that this is indeed a significant issue. Consequently, we have added a new PDF that elucidates: how to define causal representation, how to define causal consistency, and why Indefinite Data leads to such drastic inconsistency. Please refer to our Official Comment: "We add a PDF file into Supplementary Material for some Major Revisions."
>
> 2. About Hypothesis 2:
>
> We fully agree with you, and in fact, we just wanted to illustrate a potential occurrence: the possibility of interconnections among pixels. Therefore, we believe it might be more suitable to revise our paper's statement to "meaning that the representation is probably causally entangled over dimensions". Additionally, Hypothesis 2 seeks to convey that we can grasp the information of Indefinite Data without sampling. For instance, if we want to understand the climate in Singapore, we need an adequate temperature sampling to learn about the weather characteristics, such as day-night temperature difference, or whether it's a tropical or subtropical climate. However, if we wanted to comprehend the meaning of a sentence, we would not need to collect many samples of this sentence. We could understand its semantics with just one "sample", as different dimensions in the multi-value representation could be causally entangled, thus containing sufficient information.
>
> 3. About celebA:
>
> We are delighted to discuss these issues with you. Intervention is determined based on how causal variables are identified. For example, if we believe that a particular part (like background) causes bias in the representation, then the 'background' should be a parent node of the representation. It should be removed in our framework (of course, the causal relationships cannot be observed with only these two variables). This is somewhat similar to finding shortcuts and causal patterns in XAI areas. Furthermore, if we think a person's recognition result is influenced by some people's face photos, we should sequentially intervene on different parent nodes (representations of other influential people) for different people.
>
> Once again, thank you for your careful reading, and we apologize for the unclear areas in our previous manuscript. We look forward to your reply.

---

> > ### Comment · Reviewer_2DS3 · 2023-11-18
> >
> > I think most of my concerns have been addressed or will be addressed in the revised version. Thank you for your reply.

---

### Official Review · Reviewer_cecZ · 2023-10-30

**Soundness:** 3 good
**Presentation:** 4 excellent
**Contribution:** 3 good
**Rating:** 5
**Confidence:** 4

**Summary:**

The paper performs research on the intersection of deep learning and causal discovery. By so-called causal consistency, it proposes a self-supervised learning framework which provides potential support for the LLM. with some test results on a new dataset.

**Strengths:**

1. The idea is novel, and the viewpoint of consistency is important.
2. The proposed new dataset may serve the community.

**Weaknesses:**

1. Some definitions need to be more accurate.
2. Experimental section can be improved.

**Questions:**

1. The definition of "non-statistical data" forms like images, videos, and text is vague. These data still contains statistical information, and what do you mean by "non-statistical"?
2. Definition 1 (Causal Data). I still think the definition of "Indefinite Data" "Semi-definite" based on D and M only is weak. How to quantify causal consistency is still an open question. This also applies to section 2.3.
3. Def 3. The symbols looks slightly wired.
4. Section 6. I am still puzzled that how you evaluate the "causal graph" and "structure" both. It seems that these two things basically align with the same aspect of the algorithm. Why not evaluate "variance of learning results" or ”scalability of the algorithm“？

---

> ### Author Response · Authors · 2023-11-11
> **Response to reviewer cecZ**
>
> Thank you for your meticulous review. We will answer your questions in order, hoping to eliminate your concerns:
> 1. About Consistency:
>
> We believe your question 2 and question 4 are pointing to the same issue: How to measure or define causal consistency? Indeed, in line with your thoughts and those of the other two reviewers, a few weeks ago, we also realized that the introduction to causal consistency in our paper was as unappetizing as garbage. Consequently, we uploaded a file that re-interprets what causal representation is, how to measure causal consistency, and why Indefinite Data would lead to such intense causal inconsistency. We hope these changes will make it more..."delicious" to you! Please refer to our Official Comment "We add a PDF file into Supplementary Material for some Major Revisions." We trust that after understanding these supplements, you will see why our experiments need to revolve around causal graphs (causal structure) and causal representation. If there are still any doubts, please tell us.
>
> 2. About "non-statistical":
>
> We can see why you might have misunderstood, so we think it would be better to drop this adjective. In fact, what we wanted to express is that data like text, images, etc., are harder to exploit statistical characteristics (for example, we would not calculate high-order statistics between two sentences) compared to traditional causal domain data, but this is not important. In addition, if you want to understand more about the definition of Semi-Definite Data and Indefinite Data, you can refer to the review [1]. To make this definition strong enough, a lot of definitions and analyses must be introduced, such as "Causal Variables and Causal Representation", "Single-value and Multi-value Variables," etc. Therefore, in this paper, we only hoped to explain Causal Consistency clearly. We would be very grateful if you could accept the definition of data paradigm as reasonable.
>
> 3. The weird symbol:
>
> You are likely referring to the index of the causal partial order $\natural$ (music symbol). We will change it to $\Xi$ (greek symbol).
>
> Once again, thank you for your careful reading, and we apologize for the unclear areas in our previous manuscript. We look forward to your reply.
>
> [1]A Review and Roadmap of Deep Causal Model from Different Causal Structures and Representations. Arxiv2023.

---

> > ### Comment · Reviewer_cecZ · 2023-11-22
> > **Thank you**
> >
> > Thanks for the rebuttal. I would like to see more experimental results as raised in my Q4 before I raise my score.

---

> > > ### Author Response · Authors · 2023-11-22
> > > **New Experimental Results and Explaining**
> > >
> > > We have included our new experimental results and analysis in the updated PDF, which can be downloaded from the Supplementary Material (they are not added to the manuscript). We will present solutions to each of the experiments you raised individually.
> > >
> > > 1, Why evaluate "graph" and "structure":
> > >
> > > We sincerely apologize for this serious error on our part. You probably saw "graph" and "structure" mentioned in Tables 3 and 4. However, the correct terms should be "structure" and "representation". We realized that it was this mistake that caused your major confusion about our evaluation. Since this error is critical, we have made corrections immediately in the manuscript. With the correct terms "structure" and "representation", it becomes easy to understand- our motivation is to resolve inconsistencies in structure and representation, hence the experiment must evaluate both sides ("structure" and "representation" instead of "graph" and "structure"). As for the detailed evaluation methods, the causal structure used AUROC and Hamming distance, while the causal representation used AUROC and F1 score - these are well-established metrics. We have already explained how to evaluate inconsistencies in our previous rebuttal. You can refer to A1 in the previous round of the rebuttal and Section 2 in the revision PDF.
> > >
> > > 2, Variance of Results:
> > >
> > > We have added boxplots in Section 4 of the revised PDF to demonstrate the variance comparison intuitively. Please download the PDF from the Supplementary Material for review.
> > >
> > > 3, Scalability:
> > >
> > > We have added results of different training set scales in Section 5 of the revision PDF (Table 100). Please download the PDF from the Supplementary Material for review. Moreover, we remind you that in Section 6.4 of the manuscript, we conducted evaluations for three downstream tasks, which not only demonstrated scalability but also generality.
> > >
> > > Finally, we once again apologize for our serious error. Given the overlength of this paper, such grave mistakes can lead to unexpected confusion. We also now understand why you think that our experiment was not perfect. We deeply regret the additional time and energy impacts this has caused. We hope that this response will fully resolve any confusion and dissatisfaction with regard to the experiments.

---

### Official Review · Reviewer_dzLx · 2023-11-06

**Soundness:** 2 fair
**Presentation:** 1 poor
**Contribution:** 3 good
**Rating:** 3
**Confidence:** 3

**Summary:**

The paper proposes a causally motivated self supervised learning framework. In particular, the paper focuses on indefinite data, which refers to data that requires deep networks to represent them like text, videos, images etc. The authors show that current methods do not learn consistent structure and representations from a causal perspective, while their proposed method based on causal consistency outperforms existing SOTA methods on different benchmarks

**Strengths:**

The idea of considering consistency between representation and structure as a way of learning causal relations in an unsupervised manner is interesting.

**Weaknesses:**

I feel the paper is poorly written. Without the Appendix, the technical contributions of the paper is very difficult to understand. Ideally, the paper should be complete with the Appendix dedicated to extra information that complements the contents of the paper. The experimental setup is not clearly explained which makes appreciating them difficult. More details in questions.

**Questions:**

1. The paper studies the inconsistency problem between causal structure and representation. However, what is structure and representation is never formally defined in the paper. I understand from Definition 1, structure refers to the graph and representation is simply the output of a deep network operating on the data, like word embeddings of text. However, the authors consider two causal models one with the representation and one with the structure, which is confusing. A concrete example with the same data, but the two different U and V causal models for the structure and the representation would go a long way in making the presentation more clear.

2. Hypothesis 2 is not clear. What does the notation E(\hat{x}_{s,m,n}) = x_{s,m,n} mean? The E operator has not been defined, is it the expectation? If so, what is the expectation over?

3. Section 2.2 is incomprehensible. What is causal consistency has not been defined so far, What is been plotted in Figure 1. The caption says it is the MSE between similarity matrices of representation and structure. What are these similarity matrices? Appendix A.3 does not give these details. What is the reconstruction loss the authors are referring to here? For example, lines 130-133, the authors say in the M > 1 or D > 1 case, the optimization of causal strength f changes to a weighted linear combination of f_m for the different M structures. However, this optimization problem has never been defined. Numerous such issues plague the readability of the paper.

4. Eq 3 is unclear. What is L_k, what is being optimized? I understand at a high level we have 2 causal models, we intervene on both of them, and then have a way of checking the consistency of the two models. But beyond this high-level details, exact specifics of how the authors carry out the interventions and check for consistency is not present in the paper.

5. The authors propose a Causalogue dataset for causal discovery. However, it is not clear how this dataset was construcuted. The appendix gives some details. However, the authors handcraft 10 causal structures. It is not clear how GPT-4 was prompted to respect this causal structure while generating the utterances.

I think the entire paper needs significant restructuring to make the presentation clear and allow readers to understand the main contributions and needs more details to reproduce the results.

---

> ### Author Response · Authors · 2023-11-11
> **Response to Reviewer dzLx**
>
> Thank you for your attention to our paper. Simultaneously, we understand that under the situation of a heavy review load, it is inevitable that some key points might be overlooked when faced with our lengthy manuscript. Therefore, we hope that through our discussion, we can reach a pleasant consensus. I will reply to your questions one by one:
>
> 1. More Information about Causal Inconsistency:
>
> We completely agree with you. There, we added content to explain how causal consistency is measured and definitions of causal representation, etc. Please refer to our Official Comment: "We add a PDF file into Supplementary Material for some Major Revisions."
>
> 2. Question about Hypothesis 2:
>
> Yes, $E$ stands for expectation. $E(\hat{x_{s,m,n}}) \doteq x_{s,m,n}$ means that if there are many identically distributed samples of the causal representation $\hat{x_{s,m,n}}$, their expectation could equally reflect the corresponding causal variable $x_{s,m,n}$. This is the same as what we said in the lines 110-111: "For instance, any sentence is enough to express its semantics, a single image can be read for its content." For example, for the sentence "good morning", if we collect different causal representations of this sentence from different models, given a large enough number, its expected representation represents the semantics of the sentence itself (therefore, we didn't use $=$ but $\doteq$).
>
> 3. Why Causal Inconsistency Arises:
>
> We have revised Section 2.2, please refer to our Official Comment for details. Moreover, because everyone regards the generation process of causal representation as a general generation model, reconstruction loss, and optimization would emerge. For instance, section VI a) in the authoritative paper "Towards Causal Representation Learning".
>
> 4. About Eqn3:
>
> There was a typo here, it should not be $\mathcal{L}_{k}$ but $\mathcal{L}$ because the correct optimization function should not contain parameters unique to the intervention $k$. The optimization aim is the entire self-supervision process, so it involves output and parameters. Additionally, you perhaps overlooked that we proposed two implementations in Section 4.2 to assist readers in understanding our SSL framework, the details of these two implementations are in Appendix C.
>
> 5. About Causalogue Dataset:
>
> We do not primarily rely on prompts to satisfy causal structures. We set up "system" and "role" so that GPT will respect our causal structures. Please note that we are not using the window chat version, i.e., we use GPT-4 instead of ChatGPT. The details are shown in Appendix D.2 (Creation Process).
>
> Hope these explanation could solve your questions. We look forward to hear your responses. Thank you for your concerning again!

---

### Author Response · Authors · 2023-11-11
**We add a PDF file into Supplementary Material for some Major Revisions**

We gratefully acknowledge the attention of all reviewers. All reviewers requested a detailed explanation of "causal consistency". This made us realize that our explanations of some concepts might not be clear enough. Therefore, we have uploaded some points we believe need adding and modifying. You can see them in the supplementary materials. Specifically, we added the following content:
1. Explanation and definition of causal structure and causal representation.
2. How to measure causal inconsistency, or in other words, how causal inconsistency is defined.

Also, we modified the following content:
1. Section 2.2, "Why Causal Inconsistency Arises": we revised the analysis in this section and discussed the expected circumstances of causal consistency separately for the four data types. This provides a clearer explanation of why the inconsistency of Indefinite Data is significantly greater than that of other data paradigms.

If these revisions are accepted, we will incorporate them into the paper.

---

> ### Author Response · Authors · 2023-11-22
> **Second Revisions**
>
> We have updated our previous revision PDF. In this update, we have added two more sections to address reviewer cecZ's requirements on "variance" and "scalability" experimentations.

---

### Meta-Review · Area_Chair_988j · 2023-12-04

**Metareview:**

This paper proposes causal consistency conditions between causal structures and representations. Interestingly, the condition can be used as an "assumption" to learn models. Overall, I side with the reviewers, the paper is interesting and I find the angle innovative. It should definitely be resubmitted to another venue. The reason for rejection is the consistent view among reviewers that the presentation was insufficient to get the message across. While the paper's standing among reviewers has improved after the rebuttal, the question "what is the causal inconsistency between causal representation and causal structure and why do we want to optimize it" remained not sufficiently clear. I strongly encourage the authors to polish their paper to make sure their message gets across.

**Justification For Why Not Higher Score:**

The paper had major writing issues, making it difficult to understand.

**Justification For Why Not Lower Score:**

The paper seems technically solid.

---

### Decision · Program_Chairs · 2024-01-16

Reject